# CHINESE CHARACTER DECOMPOSITION WITH COMPOSITIONAL LATENT COMPONENTS

## ABSTRACT

Humans can decompose Chinese characters into compositional components and recombine them to recognize unseen characters. This reflects two cognitive principles: *Compositionality*, the idea that complex concepts are built on simpler parts; and *Learning-to-learn*, the ability to learn strategies for decomposing and recombining components to form new concepts. These principles provide inductive biases that support efficient generalization. They are critical to Chinese character recognition (CCR) in solving the zero-shot problem, which results from the common long-tail distribution of Chinese character datasets. Existing methods have made substantial progress in modeling compositionality via predefined radical or stroke decomposition. However, they often ignore the learning-to-learn capability, limiting their ability to generalize beyond human-defined schemes. Inspired by these principles, we propose a deep latent variable model that learns **Co**mpositional **La**tent components of Chinese characters (CoLa) without relying on human-defined decomposition schemes. Recognition and matching can be performed by comparing compositional latent components in the latent space, enabling zero-shot character recognition. The experiments illustrate that CoLa outperforms previous methods in both character the radical zero-shot CCR. Visualization indicates that the learned components can reflect the structure of characters in an interpretable way. Moreover, despite being trained on historical documents, CoLa can analyze components of oracle bone characters, highlighting its cross-dataset generalization ability.

## 1 INTRODUCTION

Humans exhibit remarkable flexibility in recognizing Chinese characters by understanding the internal structure of characters. Chinese characters are composed of components that often carry semantic or categorical cues. Skilled readers can decompose complex characters into constituent parts and generalize across structurally similar but previously unseen characters (Shu & Anderson, 1997; Chan & Nunes, 1998). Previous studies show that young children, even English-speaking children, can make informed guesses about unfamiliar Chinese characters based on components (Shu et al., 2000; 2003; Tang et al., 2024). Understanding this compositional mechanism offers critical insight into the design of AI systems for Chinese character recognition.

The principles of compositionality and learning-to-learn, widely discussed in cognitive science, have been proposed to explain how humans decompose abstract concepts into constituent parts and recombine known components to form new ones (Schyns et al., 1998; Winston & Horn, 1975; Smith et al., 2002). Compositionality refers to the idea that complex concepts are structured from simpler parts. Learning-to-learn indicates the ability to automatically acquire strategies for concept decomposition and component recombination. Psychological studies (Freyd, 1983) have demonstrated that these principles also underlie the ability to recognize unseen handwritten characters. Inspired by these cognitive mechanisms, some machine learning models (Lake et al., 2015; 2011; 2017) decompose handwritten characters into strokes and recombine them to generate new characters, enabling rapid generalization in simple handwritten characters. These findings suggest that incorporating the principles into intelligence systems is feasible to realize human-like generalization and Chinese character recognition abilities.

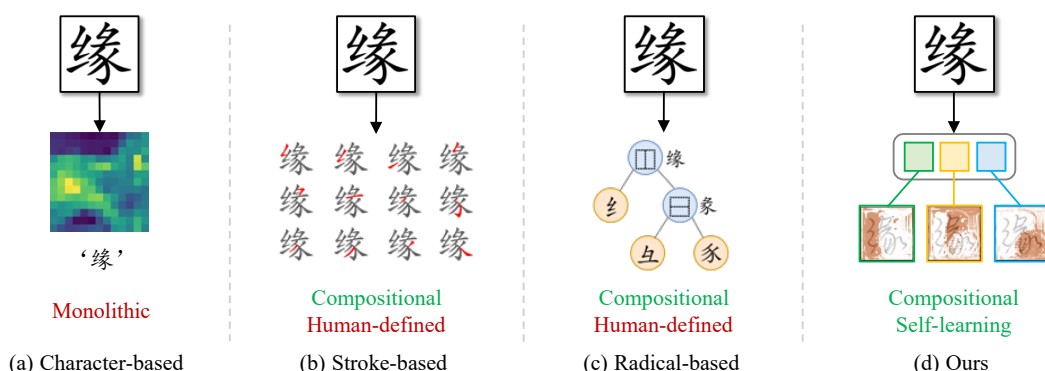

Monolithic     Compositional     Compositional     Compositional
Human-defined    Human-defined    Self-learning

(a) Character-based    (b) Stroke-based    (c) Radical-based    (d) Ours

Figure 1: **Different types of Chinese character recognition methods.** (a) character-based methods extract monolithic representations for prediction; (b) and (c) are stroke-based and radical-based methods requiring human-defined decomposition schemes to predict stroke and radical sequences; (d) the proposed CoLa automatically decomposes characters into compositional components.

Due to the large character set, Chinese character datasets typically follow a long-tail distribution. As a result, the zero-shot recognition problem, where test characters are absent from the training set, is inevitable in practical scenarios, such as the digitalization of Chinese historical documents. The above principles provide inductive biases that support efficient generalization and are critical to Chinese character recognition (CCR) in solving the zero-shot problem. The existing approaches have made substantial progress in modeling compositionality by decomposing characters via predefined schemes. They usually rely on human-defined decomposition rules, for example, as shown in Figure 1, the stroke- and radical-based approaches (Wang et al., 2019; 2018; Chen et al., 2021) utilize an auto-regressive decoder to predict corresponding stroke or radical sequences. Recently, based on CLIP (Radford et al., 2021), Yu et al. (2023) proposed an efficient image-IDS matching method for zero-shot CCR. However, they often ignore the learning-to-learn capability to automatically acquire decomposition and recombination strategies from data, limiting their ability to generalize beyond predefined decomposition schemes.

Inspired by the compositionality and learning-to-learn principles of human cognition, we propose a deep latent variable model to learn **Co**mpositional **La**tent components from Chinese characters (CoLa). CoLa decomposes Chinese characters into latent compositional components without relying on predefined decomposition schemes such as radicals or strokes. CoLa encodes an input image into component-specific representations in a latent space, which are decoded and recombined to reconstruct the visual features of the input. CoLa compares the input image and templates to determine the most likely character class based on the similarity of compositional latent components, which enables zero-shot recognition of Chinese characters. In our experiments, CoLa significantly outperforms previous methods in the radical zero-shot setting. Visualization results further demonstrate that the components learned by CoLa capture the structure of Chinese characters in an interpretable manner. Although trained on historical documents, CoLa can generalize effectively to decompose and match latent components of oracle bone characters, Korean, and Japanese, highlighting its cross-dataset generalization ability.

## 2 RELATED WORKS

### 2.1 ZERO-SHOT CHINESE CHARACTER RECOGNITION

Due to the significantly larger number of Chinese characters compared to Latin characters, character recognition in Chinese inevitably encounters zero-shot problems, *i.e.*, the characters in the test set are excluded in the training set. Early works in Chinese character recognition can be broadly categorized into three types: **1) Character-based.** Before the era of deep learning, the character-based methods usually utilize the hand-crafted features to represent Chinese characters (Jin et al., 2001; Su & Wang, 2003; Chang, 2006). With deep learning achieving a great success, MCDNN (Cireşan & Meier, 2015) takes the first attempt to use CNN for extracting robust features of Chinese characters while

approaching the human performance on handwritten CCR in the ICDAR 2013 competition (Yin et al., 2013). **2) Radical-based.** To solve the character zero-shot problem, some methods propose to predict the radical sequence of the input character image. In (Wang et al., 2018), character images are first fed into a DenseNet-based encoder (Huang et al., 2017) to extract the character features, which are subsequently decoded into the corresponding radical sequences through an attention-based decoder. However, the prediction of radical sequences takes longer time than the character-based methods. Although HDE (Cao et al., 2020) adopts a matching-based method to avoid the time-consuming radical sequence prediction, this method needs to manually design a unique vector for each Chinese character. **3) Stroke-based.** To fundamentally solve the zero-shot problem, some methods decompose Chinese characters into stroke sequences. The early stroke-based methods usually extract strokes by traditional strategies. For example, in (Kim et al., 1999), the authors employed mathematical morphology to extract each stroke in characters. The proposed method in (Liu et al., 2001) describes each Chinese character as an attributed relational graph. Recently, a deep-learning-based method (Chen et al., 2021) is proposed to decompose each Chinese character into a sequence of strokes and employs a feature-matching strategy to solve the one-to-many problem (*i.e.*, there is a one-to-many relationship between stroke sequences and Chinese characters).

Recently, Yu et al. (2023) introduced CCR-CLIP, which aligns character images with their radical sequences to recognize zero-shot characters, achieving comparable inference efficiency with the character-based approach. All previous methods focus on learning Chinese character features through human-defined representations but struggle to achieve high generalization capabilities.

## 2.2 Object-centric Representation Learning

Object-centric representation methods interpret the world in terms of objects and their relationships. They capture structured representations that are more interpretable, compositional, and generalizable, which has become increasingly popular in computer vision, as it aligns with how humans perceive and interact with the world. One class of models extracts object-centric representations with feedforward processes. For example, SPACE and GNM (Lin et al., 2020; Jiang & Ahn, 2020) attempt to divide images into small patches for parallel computation while modeling layouts of scenes. Another class of models initializes and updates object-centric representations by iterative processes (Greff et al., 2017; 2019; Emami et al., 2021). A representative method is Slot Attention, which assigns visual features to initialized slots via iterative cross-attention mechanism (Locatello et al., 2020). Based on Slot Attention, many methods have been proposed to improve the quality of object-centric representations in different scenarios (Seitzer et al., 2022).

## 3 Methodology

In this section, we introduce CoLa, a deep latent variable model that learns compositional components from Chinese characters. CoLa encodes characters into latent compositional representations without relying on radical- or stroke-level annotations, nor alignment with predefined standards. In the following sections, we introduce CoLa in detail.

## 3.1 Overall Framework

We denote the input character image as $\boldsymbol{X}$ and its class label as $y$, where $y \in \mathcal{C}$ and $\mathcal{C}$ is the set of all class labels. $N$ template character images are provided for each character in $\mathcal{C}$. The set of all template images is $\mathcal{T}$, where $\mathcal{T}_{ij}$ indicates the $j$-th template image of the $i$-th character in $\mathcal{C}$. The templates are generated from public font files and matched to predict the class $y$.

Given templates $\mathcal{T}$ and the input image $\boldsymbol{X}$, a traditional CCR method predicts the label of $\boldsymbol{X}$ by maximizing $p(y|\boldsymbol{X}, \mathcal{T})$. Unlike the traditional methods, CoLa decomposes a character image into *compositional latent components* to extract individual representations. Besides the label prediction task, CoLa incorporates another feature reconstruction objective to ensure that the components retain visual information from the input image. The objective of CoLa is to maximize

$$p(\boldsymbol{F}, y|\boldsymbol{X}, \mathcal{T}, \boldsymbol{\epsilon}) = \int p(\boldsymbol{F}, y, \boldsymbol{S}, \boldsymbol{T}|\boldsymbol{X}, \mathcal{T}, \boldsymbol{\epsilon})\mathrm{d}\boldsymbol{S}\mathrm{d}\boldsymbol{T}, \tag{1}$$

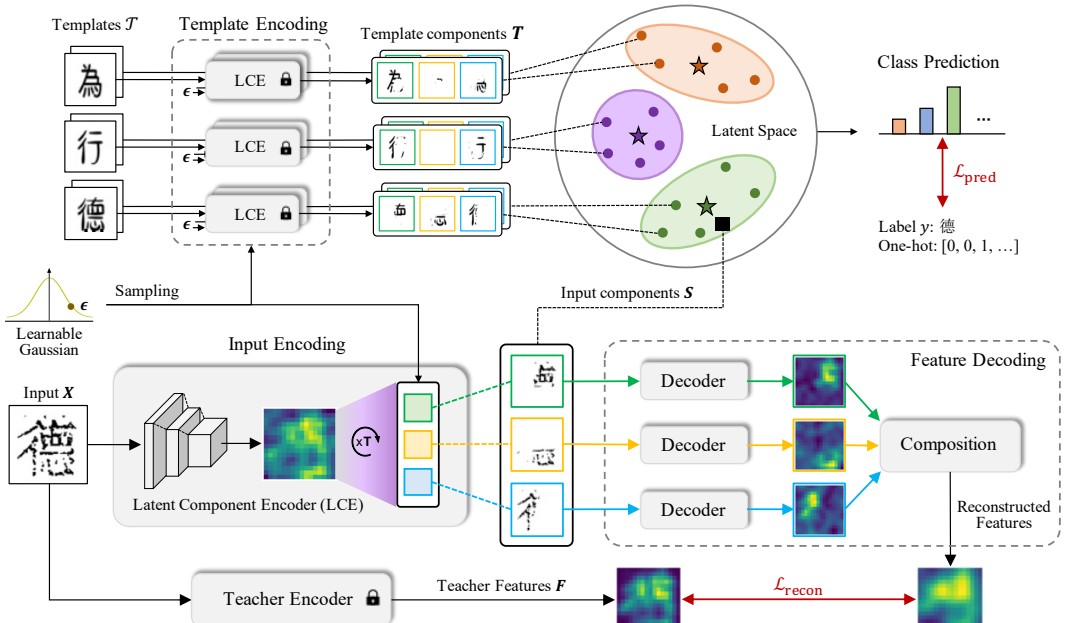

Figure 2: **The overview of CoLa.** CoLa extract compositional latent components of the input image and template images in the input encoding anf feature decoding processes, respectively. The input compositional latent components are decoded to reconstruct teacher features. The template compositional latent components constitute a mixture of Gaussians in the latent space, which is used to predict the class of the input image. The training objective is to minimize the distance between the reconstructed features and the features from a frozen teacher encoder, while maximizing the probability that CoLa predicts the correct class label of the input image.

where $\boldsymbol{F}$ denotes the visual features of $\boldsymbol{X}$, and $\boldsymbol{S}$ and $\boldsymbol{T}$ represent the compositional latent components of $\boldsymbol{X}$ and the template set $\mathcal{T}$, respectively. Although $\boldsymbol{S}$ and $\boldsymbol{T}$ appear as intermediate variables in Equation 1, they are treated as latent variables and marginalized. This is because they are neither directly observed nor explicitly defined like radicals or strokes. Instead, they gradually emerge during training, guided by the visual reconstruction and classification objectives, without relying on any external supervision. $\boldsymbol{S}$ and $\boldsymbol{T}$ are permutation invariant, i.e., using different component orders will not alter the represented character. To model the permutation invariance, CoLa introduces an observed variable $\boldsymbol{\epsilon}$ that specifies the order of components. In the following sections, we describe how CoLa decomposes the conditional generative process $p(\boldsymbol{F}, y, \boldsymbol{S}, \boldsymbol{T} | \boldsymbol{X}, \boldsymbol{\epsilon}, \mathcal{T})$ in Equation 1 into a set of distinct subprocesses and optimizes the model parameters.

## 3.2 CONDITIONAL GENERATIVE PROCESS

CoLa decomposes the conditional generative process into four processes:

$$p(\boldsymbol{F}, y, \boldsymbol{S}, \boldsymbol{T} | \boldsymbol{X}, \boldsymbol{\epsilon}, \mathcal{T}) = \underbrace{p(\boldsymbol{S}|\boldsymbol{X}, \boldsymbol{\epsilon})}_{\text{Input Encoding}} \cdot \underbrace{p(\boldsymbol{T}|\mathcal{T}, \boldsymbol{\epsilon})}_{\text{Template Encoding}} \cdot \underbrace{p(\boldsymbol{F}|\boldsymbol{S})}_{\text{Feature Decoding}} \cdot \underbrace{p(y|\boldsymbol{S}, \boldsymbol{T})}_{\text{Class Prediction}}. \quad (2)$$

CoLa employs an encoder to extract compositional latent components from input images, followed by a decoder to reconstruct visual features based on the representations. The compositional latent components of template images are also extracted and used to predict the class of the input image in the latent space. The generative process of CoLa is composed of the following parts.

**Input Encoding.** We define $p(\boldsymbol{S}|\boldsymbol{X}, \boldsymbol{\epsilon})$ as a Gaussian distribution $\mathcal{N}(\boldsymbol{\mu}^s, \sigma^2 \boldsymbol{I})$, where $\sigma$ is a hyper-parameter. A Latent Component Encoder (LCE) to introduced to estimate $\boldsymbol{\mu}^s \in \mathbb{R}^{K \times D}$, where $K$ is the maximum number of components extracted from the images. $\boldsymbol{X}$ is encoded by a CNN-based backbone, augmented with positional embeddings, and transformed into visual features through layer normalization and a multilayer Perceptron. We denote the visual features as $\boldsymbol{H} \in \mathbb{R}^{M \times D}$

where $M$ is the number of input features and $D$ is the dimensionality of each feature. $\boldsymbol{\mu}^s$ is initialized by $\boldsymbol{\epsilon}$. In the following iterations, LCE measures the similarity between components and visual features, updating the components through the slot attention mechanism (Locatello et al., 2020), which allows each component to gradually focus on different regions of $\boldsymbol{X}$. Controlling $K$ encourages LCE to learn interpretable components rather than decomposing the input into more fragmented parts. Finally, we sample the input compositional latent components by $\boldsymbol{S} \sim \mathcal{N}(\boldsymbol{\mu}^s, \sigma^2 \boldsymbol{I})$.

**Template Encoding.** The template encoding process is factorized in terms of each template image:

$$p(\boldsymbol{T}|\mathcal{T}, \boldsymbol{\epsilon}) = \prod_{i \in \mathcal{C}} \prod_{n=1}^{N} p(\boldsymbol{T}_{i,n}|\mathcal{T}_{i,n}, \boldsymbol{\epsilon}) = \prod_{i \in \mathcal{C}} \prod_{n=1}^{N} \mathcal{N}(\boldsymbol{\mu}_{i,n}^t, \sigma^2 \boldsymbol{I}), \quad (3)$$

where $\boldsymbol{\mu}_{i,n}^t$ of each template image is estimated using LCE. The template compositional latent components are separately sampled via $\boldsymbol{T}_{i,n} \sim \mathcal{N}(\boldsymbol{\mu}_{i,n}^t, \sigma^2 \boldsymbol{I})$ for $i \in \mathcal{C}$ and $n = 1, \cdots, N$.

**Feature Decoding.** The decoder of CoLa transforms $\boldsymbol{S}$ into the input features $\boldsymbol{F}$ extract by a teacher encoder. $\boldsymbol{H}$ is not chosen as the reconstruction target, since the backbone is updated during training, which may lead the model to exploit shortcuts that minimize reconstruction loss, *e.g.*, collapsing $\boldsymbol{H}$ to a zero matrix. On the other hand, $\boldsymbol{H}$ may contain low-level information to reconstruct image details, while neglecting high-level semantics that are more useful for recognition tasks. Seitzer et al. (Seitzer et al., 2022) point out that well-pretrained visual features can facilitate the model in learning components that make up images. Inspired by this idea, CoLa introduces a pretrained teacher encoder, aligning the output of the decoder and the teacher encoder to enhance the ability of learning compositional latent components. The teacher encoder consists of a frozen DINOv2 encoder (Oquab et al., 2023) and a two-layer convolutional network. The visual features extracted by the teacher are passed through a prediction head, which is jointly trained with the convolutional layers by predicting the classes of character images in the training data. The decoding process $p(\boldsymbol{F}|\boldsymbol{S})$ is formulated as $\mathcal{N}(\boldsymbol{\mu}^d, \sigma^2 \boldsymbol{I})$. CoLa uses a spatial broadcast decoder (Watters et al., 2019) to convert $\boldsymbol{S}_k$ to the component features $\boldsymbol{O}_k$ and corresponding mask $\boldsymbol{M}_k$:

$$\boldsymbol{M}_k = \frac{e^{\boldsymbol{\Lambda}_k}}{\sum_{l=1}^{K} e^{\boldsymbol{\Lambda}_l}}, \quad \text{where } \boldsymbol{\Lambda}_k, \boldsymbol{O}_k = \text{Decoder}\left(\boldsymbol{S}_k\right), \quad k = 1, \cdots, K. \quad (4)$$

$\boldsymbol{\Lambda}_k$ contains unnormalized logits that indicate where the $k$-th component contributes in the image, and is converted to the normalized mask $\boldsymbol{M}_k$. The component features are combined via mask-weighted summation, followed by a linear layer to predict the mean $\boldsymbol{\mu}^d = \text{Linear}\left(\sum_k \boldsymbol{M}_k \odot \boldsymbol{O}_k\right)$.

**Class Prediction.** The input and template images are compared according to $\boldsymbol{S}$ and $\boldsymbol{T}$ in the latent space to predict class labels. CoLa assumes that the probability of assigning $\boldsymbol{S}$ to class $i$ is determined by its distance to the center of the corresponding templates. Each class $i$ is represented by the mean of the template compositional latent components, i.e., $\bar{\boldsymbol{T}}_i = \sum_{n=1}^{N} \boldsymbol{T}_{i,n}/N$. Therefore, $p(y|\boldsymbol{S}, \boldsymbol{T})$ is modeled as a categorical distribution, where

$$p(y = i|\boldsymbol{S}, \boldsymbol{T}) \propto \exp\left(-\left\|\boldsymbol{S} - \bar{\boldsymbol{T}}_i\right\|_2^2\right). \quad (5)$$

### 3.3 Parameter Learning

The objective in Equation 1 is typically intractable when the conditional generative process is parameterized with neural networks, since we need to estimate the integration over $\boldsymbol{S}$ and $\boldsymbol{T}$. The stochastic gradient variational Bayes (SGVB) estimator (Kingma & Welling, 2013; Sohn et al., 2015) is applied to make the objective tractable by estimating the log-form of the likelihood through the evidence lower bound (ELBO). The core idea is to approximate the posterior distribution $p(\boldsymbol{S}, \boldsymbol{T}|\boldsymbol{X}, \mathcal{T}, \boldsymbol{\epsilon})$ with a variational distribution $q(\boldsymbol{S}, \boldsymbol{T}|\boldsymbol{X}, \mathcal{T}, \boldsymbol{F}, y, \boldsymbol{\epsilon})$ parameterized by neural networks. Then Equation 1 is estimated via the following lower bound (see Appendix A for the detailed derivation):

$$\text{ELBO} = \mathbb{E}_{q(\boldsymbol{S}, \boldsymbol{T}|\boldsymbol{X}, \mathcal{T}, \boldsymbol{F}, y, \boldsymbol{\epsilon})} \left[\log \frac{p(\boldsymbol{F}, y, \boldsymbol{S}, \boldsymbol{T}|\boldsymbol{X}, \mathcal{T}, \boldsymbol{\epsilon})}{q(\boldsymbol{S}, \boldsymbol{T}|\boldsymbol{X}, \mathcal{T}, \boldsymbol{F}, y, \boldsymbol{\epsilon})}\right]. \quad (6)$$

We use a parameter-shared variational distribution as an approximation of the posterior:

$$q(\boldsymbol{S}, \boldsymbol{T}|\boldsymbol{X}, \boldsymbol{\epsilon}, \mathcal{T}, \boldsymbol{F}, y) = q(\boldsymbol{S}|\boldsymbol{X}, \boldsymbol{\epsilon})q(\boldsymbol{T}|\mathcal{T}, \boldsymbol{\epsilon}), \quad (7)$$

which take the same forms as in the generative process to extract compositional latent components from the input and templates. Therefore, $q(\boldsymbol{S}|\boldsymbol{X}, \boldsymbol{\epsilon})$ and $q(\boldsymbol{T}|\mathcal{T}, \boldsymbol{\epsilon})$ are estimated by LCE to reduce the parameters. Considering Equations 2 and 7, the ELBO in Equation 6 is further refined into:

$$\text{ELBO} = \underbrace{\mathbb{E}_{q(\boldsymbol{S}|\boldsymbol{X}, \boldsymbol{\epsilon})}\Big[\log p(\boldsymbol{F}|\boldsymbol{S})\Big]}_{\text{Reconstruction Term } \mathcal{L}_{\text{recon}}} + \underbrace{\mathbb{E}_{q(\boldsymbol{S},\boldsymbol{T}|\boldsymbol{X}, \mathcal{T}, \boldsymbol{\epsilon})}\Big[\log p(y|\boldsymbol{S}, \boldsymbol{T})\Big]}_{\text{Prediction Term } \mathcal{L}_{\text{pred}}}. \tag{8}$$

The **reconstruction term** $\mathcal{L}_{\text{recon}}$ encourages the decoder to reconstruct teacher features from compositional latent components, which ensures that the compositional latent components learned by CoLa can reflect the structure of characters. The **prediction term** $\mathcal{L}_{\text{pred}}$ measures the similarity between the prediction results and ground truth class labels, which directly influences the accuracy of CCR and is critical when training CoLa to recognize Chinese characters. Finally, we compute the ELBO through a Monte Carlo estimator and the training objective of CoLa is to minimize the negative ELBO. We introduce a hyperparameter $\lambda$ to control the importance of the prediction term. After training, CoLa can directly recognize zero-shot characters based on the templates of novel character sets. The detailed derivations of the ELBO are provided in Appendix A.

## 4 EXPERIMENTS

In this section, we first introduce the experimental settings, including data construction and training details. Then, we show some results of conducted experiments (additional experimental results are shown in Appendix D) to illustrate the application of the compositional latent components, including visualization of compositional latent components, zero-shot CCR on three datasets, and an evaluation on oracle bone characters to validate the cross-dataset generalization ability of CoLa.

**Dataset Construction.** In this paper, we mainly conduct experiments on three datasets: HWDB1.0-1.1 (Liu et al., 2013), Printed artistic characters (Chen et al., 2021) and Historical Documents. HWDB1.0-1.1 (Liu et al., 2013) contains 2,678,424 handwritten Chinese character images with 3,881 classes, which is collected from 720 writers and covers 3,755 commonly-used Level-1 Chinese characters. Printed artistic characters (Chen et al., 2021) are generated in 105 font files and contain 394,275 samples for 3,755 Level-1 Chinese characters. The data of Historical Documents is collected from the web library. The examples of each dataset are shown in Appendix B.

**Training Details.** CoLa is trained using the Adam optimizer (Kingma & Ba, 2014) where the momentums $\beta_1$ and $\beta_2$ are set to 0.9 and 0.99. For the CNN-based backbone and slot attention module, we increase the learning rate from 0 to $10^{-4}$ in the first 30K steps and then halve the learning rate every 250K steps. For the spatial broadcast decoder, we increase the learning rate from 0 to $3 \times 10^{-4}$ in the first 30K steps and then halve the learning rate every 250K steps. The training batch size is 32, and the input image of CoLa will be scaled to $80 \times 80$. We set $K = 3$, $N = 10$ and $\lambda = 0.01$. More details of model architecture and training settings are provided in Appendix C.

### 4.1 VISUALIZATION OF COMPOSITIONAL LATENT COMPONENTS

In this section, we visualize the compositional latent components learned by CoLa. CoLa does not rely on any manually defined decomposition scheme, nor is it designed with the objective of aligning with radicals. The role of the compositional latent components is to serve as latent representations of distinct and independent regions within a character. The components provide a self-learning structural abstraction, while remaining free from the constraints of human-designed radical systems. Nevertheless, we compare the learned components with human-defined radical-based decompositions for reference. Our key observation is that, even without any radical-level information, CoLa can still discover decompositions that align with predefined radicals in certain cases.

**Qualitative Analysis.** As shown in Figure 3, we visualize the attended regions of components on the three datasets. The visualization results reveal that each component focuses on distinct and independent regions of the character. Despite the absence of fine-grained supervision information based on radicals or strokes, the compositional latent components can still effectively distinguish meaningful regions of the character. CoLa produces component decompositions that, in some cases, align with human-defined schemes (Examples 3, 4, and 8). In the radical annotations, complex characters are usually decomposed into a large number of fine-grained elements (Examples 1 and

Figure 3: **Visualization of the compositional latent components.** For each example, the left side are compositional latent components (Cmp#1 ∼ Cmp#3), and the right side is the human-defined radical decomposition scheme of the character. The radical regions are highlighted using red boxes.

2), whereas CoLa tends to learn higher-level structures. We observe that CoLa performs different hierarchical decompositions for characters. For example, although Cmp#1 of Example 9 can be further decomposed in the same way as Example 10, CoLa chooses to stop at the level of Example 10 without proceeding to finer splits. This suggests that CoLa develops a distinct understanding of decomposition to capture the overall structure of Chinese characters.

**Quantitative Analysis.** We conduct a quantitative analysis on the attention masks of components. Since the datasets do not provide ground-truth radical masks, we select a subset of 100 different characters from HWDB and manually annotate masks based on the radical annotations. We also apply the CRF-based postprocessing method used in image segmentation (Kamnitsas et al., 2017), and compute mIoU between the component masks and the ground-truth radical masks after postprocessing (+CRF). Table 1 reports the quantitative results where CoLa outperforms Slot Attention. Applying CRF post-processing significantly improves the performance of CoLa. This indicates that the masks pro-

Table 1: **Quantitative results of the component masks.** We compute the mIoU between component masks and ground-truth radical masks.

| Models | mIoU ↑ |
|---|---|
| Slot Attention | 0.2745 |
| Slot Attention + CRF | 0.2748 |
| CoLa | 0.3478 |
| CoLa + CRF | **0.4349** |

duced by CoLa are more structurally meaningful and therefore benefit from CRF-based refinement, whereas the lower-quality masks from Slot Attention leave little room for CRF to contribute. Overall, these results validate the effectiveness of CoLa in discovering interpretable components and show that it can be further enhanced with standard post-processing techniques.

## 4.2 Results on Chinese Character Recognition

We follow (Chen et al., 2021) to construct the corresponding datasets for the character zero-shot and radical zero-shot settings. We select several radical-based methods (Wang et al., 2018; Cao et al., 2020; Luo et al., 2023; Li et al., 2024; Zhang et al., 2025), stroke-based method (Chen et al., 2021; Yu et al., 2024; Zu et al., 2022) and matching-based method (Yu et al., 2023) as the compared methods in zero-shot settings. For fair comparison, some few-shot CCR models (Li et al., 2020), which trained with additional samples from the test character set, are not considered. Moreover, since the character accuracy of character-based methods is almost zero in zero-shot settings, these methods are also not used for comparison.

**Character Zero-Shot Setting.** For the character zero-shot settings, we collect samples with labels falling in the first $m$ classes as the training set and the last $k$ classes as the test set. For the handwritten character dataset HWDB and printed artistic character dataset, $m$ ranges in $\{500, 1000, 1500, 2000, 2755\}$ and $k$ is set to 1000. We first validate the effectiveness of CoLa in the character zero-shot setting. As shown in Table 2, regardless of the handwritten or printed character dataset, the proposed CoLa outperforms previous methods by a clear margin. For instance, in the 500 HWDB character zero-shot setting, the proposed method achieves a performance improvement of about 47% compared with the previous methods.

Table 2: **Accuracy (%) of Chinese character recognition on the character and radical zero-shot setting.** CoLa outperforms the previous methods on handwritten and printed character datasets HWDB and Printed, especially with a limited training charset (with only 500 training characters). The training of CoLa does not rely on human-defined decomposition schemes, therefore the compositional latent components learned by CoLa can handle zero-shot radicals more effectively.

| Datasets | HWDB (Character Zero-shot) | | | | | Printed (Character Zero-shot) | | | | |
|---|---|---|---|---|---|---|---|---|---|---|
| | 500 | 1000 | 1500 | 2000 | 2755 | 500 | 1000 | 1500 | 2000 | 2755 |
| DenseRAN | 1.70 | 8.44 | 14.71 | 19.51 | 30.68 | 0.20 | 2.26 | 7.89 | 10.86 | 24.80 |
| HDE | 4.90 | 12.77 | 19.25 | 25.13 | 33.49 | 7.48 | 21.13 | 31.75 | 40.43 | 51.41 |
| SD | 5.60 | 13.85 | 22.88 | 25.73 | 37.91 | 7.03 | 26.22 | 48.42 | 54.86 | 65.44 |
| ACPM | 9.72 | 18.50 | 27.74 | 34.00 | 42.43 | - | - | - | - | - |
| CUE | 7.43 | 15.75 | 24.01 | 27.04 | 40.55 | - | - | - | - | - |
| SideNet | 5.10 | 16.20 | 33.80 | 44.10 | 50.30 | - | - | - | - | - |
| HierCode | 6.22 | 20.71 | 35.39 | 45.67 | 56.21 | - | - | - | - | - |
| RSST | 11.56 | 21.83 | 35.32 | 39.22 | 47.44 | 23.12 | 42.21 | 62.29 | 66.86 | 71.32 |
| CCR-CLIP | 21.79 | 42.99 | 55.86 | 62.99 | 72.98 | 23.67 | 47.57 | 60.72 | 67.34 | 76.44 |
| Ours | **68.59** | **76.58** | **79.16** | **81.16** | **82.71** | **78.10** | **85.38** | **90.32** | **93.26** | **92.70** |

| Datasets | HWDB (Radical Zero-shot) | | | | | Printed (Radical Zero-shot) | | | | |
|---|---|---|---|---|---|---|---|---|---|---|
| | 50 | 40 | 30 | 20 | 10 | 50 | 40 | 30 | 20 | 10 |
| DenseRAN | 0.21 | 0.29 | 0.25 | 0.42 | 0.69 | 0.07 | 0.16 | 0.25 | 0.78 | 1.15 |
| HDE | 3.26 | 4.29 | 6.33 | 7.64 | 9.33 | 4.85 | 6.27 | 10.02 | 12.75 | 15.25 |
| SD | 5.28 | 6.87 | 9.02 | 14.67 | 15.83 | 11.66 | 17.23 | 20.62 | 31.10 | 35.81 |
| ACPM | 4.29 | 6.20 | 7.85 | 10.36 | 12.51 | - | - | - | - | - |
| RSST | 7.94 | 11.56 | 15.13 | 15.92 | 20.21 | 13.90 | 19.45 | 26.59 | 34.11 | 38.15 |
| CCR-CLIP | 11.15 | 13.85 | 16.01 | 16.76 | 15.96 | 11.89 | 14.64 | 17.70 | 22.03 | 21.27 |
| -pred -teach | 10.59 | 11.65 | 8.04 | 12.11 | 11.89 | 10.49 | 10.24 | 8.44 | 8.45 | 9.41 |
| -pred | 30.09 | 35.96 | 42.81 | 39.22 | 49.63 | 34.50 | 37.38 | 41.47 | 39.15 | 36.30 |
| Ours | **70.40** | **74.80** | **77.01** | **80.64** | **75.78** | **82.23** | **84.48** | **82.20** | **92.12** | **94.81** |

**Radical Zero-Shot Setting.** For the radical zero-shot settings, we first calculate the frequency of each radical in the lexicon. Then the samples of characters that have one or more radicals appearing less than $n$ times are collected as the test set, otherwise, collected as the training set, where $n$ ranges in $\{10, 20, 30, 40, 50\}$ in the radical zero-shot settings. The experimental results shown in Table 2 indicate that the proposed method achieves the best performance across all sub-settings with an average improvement of about 60% in accuracy compared to the previous methods. Since we do not introduce manually defined radical or stroke sequences for supervision, CoLa can still achieve satisfying performance in the case of radical zero-shot scenarios. Although the radicals in test sets are not common in training sets, the proposed CoLa can robustly decompose corresponding latent components, improving the zero-shot recognition performance.

**Historical Document Characters.** We also collect a dataset from historical documents to validate the effectiveness of our method. Historical documents contain more characters with complex structures, and they often exhibit broken or blurred strokes. Through the results in Table 3, we observe that the proposed CoLa can achieve a performance improvement of about 35% compared with the previous method CCR-CLIP (Li et al., 2020), which validates the robustness of CoLa in more complicated settings.

Table 3: Accuracy (%) of Chinese character recognition on historical document characters.

| Models | Accuracy ↑ |
|---|---|
| SD | 11.09 |
| DenseRAN | 13.43 |
| CCR-CLIP | 22.36 |
| Ours | **57.37** |

**Ablation Study.** The results in Table 2 show that removing the teacher encoder (-teach) or the prediction loss (-pred) leads to significant performance decreases. The teacher encoder is crucial for learning latent components from character images. Pixel-level reconstruction makes it difficult for CoLa to segment components that generalize well, while high-level teacher features can provide clearer guidance for component learning. The prediction loss aligns the latent component representations of handwritten characters with those of printed templates. In HWDB, it encourages char-

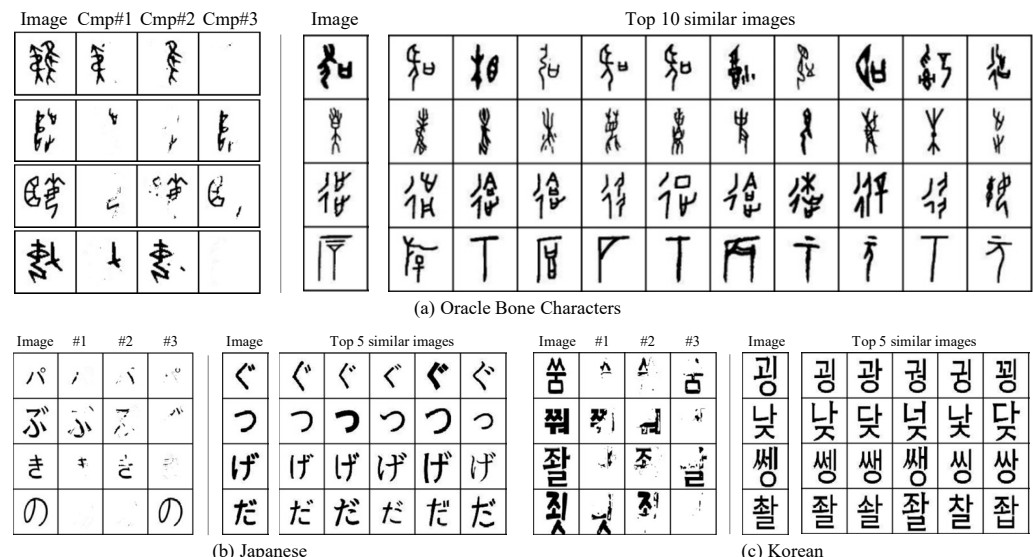

Figure 4: **Cross-dataset evaluation on Oracle Bone Characters (OBCs), Japanese characters and Korean characters.** On the left of each figure, we present the component parsing results obtained by applying CoLa trained on Historical Documents to OBCs, Japanese, and Korean. On the right of each figure, we select four examples and visualize the top-10 or top-5 similar samples.

acters belonging to the same class to be projected into similar regions of the latent space, thereby enhancing the model's ability to match components effectively and improving recognition accuracy.

## 4.3 CROSS-DATASET EVALUATION

This experiment evaluates the cross-dataset generalization ability of CoLa. We examine whether a model trained on historical Chinese characters can transfer its decomposition capability to other types of characters without retraining. To this end, we introduce three datasets, including Oracle Bone Characters (OBCs) (Wang et al., 2024), Japanese characters, and Korean characters. Figure 4 presents the decomposition results and the top similar images retrieved based on the similarity of compositional latent components. Despite the substantial visual gap between modern and OBCs, CoLa can extract components and retrieve visually related samples. For Japanese and Korean characters, CoLa can parse distinct components and retrieve similar samples, though parsing Korean characters proves more challenging due to the larger set of visually similar components. These findings confirm that CoLa demonstrates effective cross-dataset generalization and has the potential to discover compositional structures in previously unseen writing systems.

## 5 DISCUSSION

In this paper, we propose a deep latent variable model to learn Compositional Latent components of Chinese characters (CoLa) to address challenges in Chinese character recognition, particularly zero-shot recognition. CoLa offers a unique solution by automatically learning compositional components from the data as latent variables, distinct from traditional radical or stroke-based approaches. The experimental results demonstrate that CoLa outperforms previous methods in character and radical zero-shot settings. The visualization experiments also reveal that the acquired components reflect the structure of Chinese characters in an interpretable manner and can be applied to analyze oracle bone characters, Japanese characters, and Korean characters.

**Limitation.** Although CoLa achieves outperforming results in zero-shot settings, its capability in scene images with complex backgrounds or low resolution remains underexplored. The complex backgrounds and noise in scene images are unfavorable factors that impact the decomposition of components, which can be a significant topic in future work.

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

# A PROOFS AND DERIVATIONS

## A.1 ELBO

According to the stochastic gradient variational Bayes (Kingma & Welling, 2013; Sohn et al., 2015), the log-likelihood $p(\boldsymbol{F}, y|\boldsymbol{X}, \boldsymbol{\epsilon}, \mathcal{T})$ can be estimated using the following evidence lower bound (ELBO).

$$
\begin{aligned}
& \log p(\boldsymbol{F}, y|\boldsymbol{X}, \boldsymbol{\epsilon}, \mathcal{T}) \\
& = \int q(\boldsymbol{S}, \boldsymbol{T}|\boldsymbol{X}, \mathcal{T}, \boldsymbol{\epsilon}, \boldsymbol{F}, y) \log p(\boldsymbol{F}, y|\boldsymbol{X}, \boldsymbol{\epsilon}, \mathcal{T}) \mathrm{d}\boldsymbol{S}\mathrm{d}\boldsymbol{T} \\
& = \int q(\boldsymbol{S}, \boldsymbol{T}|\boldsymbol{X}, \mathcal{T}, \boldsymbol{\epsilon}, \boldsymbol{F}, y) \log \frac{p(\boldsymbol{F}, y, \boldsymbol{S}, \boldsymbol{T}|\boldsymbol{X}, \boldsymbol{\epsilon}, \mathcal{T})q(\boldsymbol{S}, \boldsymbol{T}|\boldsymbol{X}, \mathcal{T}, \boldsymbol{\epsilon}, \boldsymbol{F}, y)}{p(\boldsymbol{S}, \boldsymbol{T}|\boldsymbol{F}, y, \boldsymbol{X}, \boldsymbol{\epsilon}, \mathcal{T})q(\boldsymbol{S}, \boldsymbol{T}|\boldsymbol{X}, \mathcal{T}, \boldsymbol{\epsilon}, \boldsymbol{F}, y)} \mathrm{d}\boldsymbol{S}\mathrm{d}\boldsymbol{T} \\
& = \int q(\boldsymbol{S}, \boldsymbol{T}|\boldsymbol{X}, \mathcal{T}, \boldsymbol{\epsilon}, \boldsymbol{F}, y) \log \frac{p(\boldsymbol{F}, y, \boldsymbol{S}, \boldsymbol{T}|\boldsymbol{X}, \boldsymbol{\epsilon}, \mathcal{T})}{q(\boldsymbol{S}, \boldsymbol{T}|\boldsymbol{X}, \mathcal{T}, \boldsymbol{\epsilon}, \boldsymbol{F}, y)} \mathrm{d}\boldsymbol{S}\mathrm{d}\boldsymbol{T} \\
& \quad + \int q(\boldsymbol{S}, \boldsymbol{T}|\boldsymbol{X}, \mathcal{T}, \boldsymbol{\epsilon}, \boldsymbol{F}, y) \log \frac{q(\boldsymbol{S}, \boldsymbol{T}|\boldsymbol{X}, \mathcal{T}, \boldsymbol{\epsilon}, \boldsymbol{F}, y)}{p(\boldsymbol{S}, \boldsymbol{T}|\boldsymbol{F}, y, \boldsymbol{X}, \boldsymbol{\epsilon}, \mathcal{T})} \mathrm{d}\boldsymbol{S}\mathrm{d}\boldsymbol{T} \\
& = \int q(\boldsymbol{S}, \boldsymbol{T}|\boldsymbol{X}, \mathcal{T}, \boldsymbol{\epsilon}, \boldsymbol{F}, y) \log \frac{p(\boldsymbol{F}, y, \boldsymbol{S}, \boldsymbol{T}|\boldsymbol{X}, \boldsymbol{\epsilon}, \mathcal{T})}{q(\boldsymbol{S}, \boldsymbol{T}|\boldsymbol{X}, \mathcal{T}, \boldsymbol{\epsilon}, \boldsymbol{F}, y)} \mathrm{d}\boldsymbol{S}\mathrm{d}\boldsymbol{T} \\
& \quad + \mathrm{KL}\big(q(\boldsymbol{S}, \boldsymbol{T}|\boldsymbol{X}, \mathcal{T}, \boldsymbol{\epsilon}, \boldsymbol{F}, y) \,\|\, p(\boldsymbol{S}, \boldsymbol{T}|\boldsymbol{F}, y, \boldsymbol{X}, \boldsymbol{\epsilon}, \mathcal{T})\big) \\
& \geq \mathbb{E}_{q(\boldsymbol{S}, \boldsymbol{T}|\boldsymbol{X}, \mathcal{T}, \boldsymbol{\epsilon}, \boldsymbol{F}, y)} \left[ \log \frac{p(\boldsymbol{F}, y, \boldsymbol{S}, \boldsymbol{T}|\boldsymbol{X}, \boldsymbol{\epsilon}, \mathcal{T})}{q(\boldsymbol{S}, \boldsymbol{T}|\boldsymbol{X}, \mathcal{T}, \boldsymbol{\epsilon}, \boldsymbol{F}, y)} \right] = \mathrm{ELBO}
\end{aligned}
\tag{9}
$$

Given the conditional generative process $p(\boldsymbol{F}, y, \boldsymbol{S}, \boldsymbol{T}|\boldsymbol{X}, \boldsymbol{\epsilon}, \mathcal{T})$ and the variational distribution $q(\boldsymbol{S}, \boldsymbol{T}|\boldsymbol{X}, \mathcal{T}, \boldsymbol{\epsilon}, \boldsymbol{F}, y)$ in Equations 2 and 7:

$$
\begin{aligned}
p(\boldsymbol{F}, y, \boldsymbol{S}, \boldsymbol{T}|\boldsymbol{X}, \boldsymbol{\epsilon}, \mathcal{T}) &= p(\boldsymbol{S}|\boldsymbol{X}, \boldsymbol{\epsilon})p(\boldsymbol{F}|\boldsymbol{S})p(\boldsymbol{T}|\mathcal{T}, \boldsymbol{\epsilon})p(y|\boldsymbol{S}, \boldsymbol{T}), \\
q(\boldsymbol{S}, \boldsymbol{T}|\boldsymbol{X}, \mathcal{T}, \boldsymbol{\epsilon}, \boldsymbol{F}, y) &= q(\boldsymbol{S}|\boldsymbol{X}, \boldsymbol{\epsilon})q(\boldsymbol{T}|\mathcal{T}, \boldsymbol{\epsilon}),
\end{aligned}
\tag{10}
$$

the ELBO can be further factorized via Equation 8:

$$
\begin{aligned}
\mathrm{ELBO} &= \mathbb{E}_{q(\boldsymbol{S}, \boldsymbol{T}|\boldsymbol{X}, \mathcal{T}, \boldsymbol{\epsilon}, \boldsymbol{F}, y)} \left[ \log \frac{p(\boldsymbol{S}|\boldsymbol{X}, \boldsymbol{\epsilon})p(\boldsymbol{F}|\boldsymbol{S})p(\boldsymbol{T}|\mathcal{T}, \boldsymbol{\epsilon})p(y|\boldsymbol{S}, \boldsymbol{T})}{q(\boldsymbol{S}|\boldsymbol{X}, \boldsymbol{\epsilon})q(\boldsymbol{T}|\mathcal{T}, \boldsymbol{\epsilon})} \right] \\
&= \mathbb{E}_{q(\boldsymbol{S}|\boldsymbol{X}, \boldsymbol{\epsilon})} \left[ \mathbb{E}_{q(\boldsymbol{T}|\mathcal{T}, \boldsymbol{\epsilon})} \left[ \log p(\boldsymbol{F}|\boldsymbol{S}) \right] \right] + \mathbb{E}_{q(\boldsymbol{S}|\boldsymbol{X}, \boldsymbol{\epsilon})} \left[ \mathbb{E}_{q(\boldsymbol{T}|\mathcal{T}, \boldsymbol{\epsilon})} \left[ \log p(y|\boldsymbol{S}, \boldsymbol{T}) \right] \right] \\
&\quad + \mathbb{E}_{q(\boldsymbol{S}|\boldsymbol{X}, \boldsymbol{\epsilon})} \left[ \mathbb{E}_{q(\boldsymbol{T}|\mathcal{T}, \boldsymbol{\epsilon})} \left[ \log \frac{p(\boldsymbol{S}|\boldsymbol{X}, \boldsymbol{\epsilon})}{q(\boldsymbol{S}|\boldsymbol{X}, \boldsymbol{\epsilon})} \right] \right] + \mathbb{E}_{q(\boldsymbol{S}|\boldsymbol{X}, \boldsymbol{\epsilon})} \left[ \mathbb{E}_{q(\boldsymbol{T}|\mathcal{T}, \boldsymbol{\epsilon})} \left[ \log \frac{p(\boldsymbol{T}|\mathcal{T}, \boldsymbol{\epsilon})}{q(\boldsymbol{T}|\mathcal{T}, \boldsymbol{\epsilon})} \right] \right] \\
&= \underbrace{\mathbb{E}_{q(\boldsymbol{S}|\boldsymbol{X}, \boldsymbol{\epsilon})} \left[ \log p(\boldsymbol{F}|\boldsymbol{S}) \right]}_{\text{Reconstruction Term } \mathcal{L}_{\text{recon}}} + \underbrace{\mathbb{E}_{q(\boldsymbol{S}, \boldsymbol{T}|\boldsymbol{X}, \mathcal{T}, \boldsymbol{\epsilon})} \left[ \log p(y|\boldsymbol{S}, \boldsymbol{T}) \right]}_{\text{Prediction Term } \mathcal{L}_{\text{pred}}} \\
&\quad - \underbrace{\mathrm{KL}\big(q(\boldsymbol{S}|\boldsymbol{X}, \boldsymbol{\epsilon}) \,\|\, p(\boldsymbol{S}|\boldsymbol{X}, \boldsymbol{\epsilon})\big)}_{\text{Input Regularizer } \mathcal{R}_{\text{input}}} - \underbrace{\mathrm{KL}\big(q(\boldsymbol{T}|\mathcal{T}, \boldsymbol{\epsilon}) \,\|\, p(\boldsymbol{T}|\mathcal{T}, \boldsymbol{\epsilon})\big)}_{\text{Template Regularizer } \mathcal{R}_{\text{temp}}}.
\end{aligned}
\tag{11}
$$

## A.2 DETAILS OF THE CONDITIONAL GENERATIVE PROCESS

The template encoding process is factorized via $p(\boldsymbol{T}|\mathcal{T}, \boldsymbol{\epsilon}) = \prod_{i \in \mathcal{C}} \prod_{n=1}^{N} p(\boldsymbol{T}_{i,n}|\mathcal{T}_{i,n}, \boldsymbol{\epsilon})$. The compositional latent components of the input image $\boldsymbol{X}$ and templates $\mathcal{T}$ are extracted by

$$
\begin{aligned}
\boldsymbol{S} &\sim \mathcal{N}\left(\boldsymbol{\mu}^s, \sigma^2 \boldsymbol{I}\right), \quad \text{where } \boldsymbol{\mu}^s = \mathrm{LCE}\left(\boldsymbol{X}\right), \\
\boldsymbol{T}_{i,n} &\sim \mathcal{N}\left(\boldsymbol{\mu}^t_{i,n}, \sigma^2 \boldsymbol{I}\right), \quad \text{where } \boldsymbol{\mu}^t_{i,n} = \mathrm{LCE}\left(\mathcal{T}_{i,n}\right), \quad i \in \mathcal{C}, \quad n = 1, \cdots, N.
\end{aligned}
\tag{12}
$$

Denoting the Spatial Broadcast Decoder and the composition process used in feature decoding as a function CompDec, the reconstructed teacher features are obtained via

$$
\tilde{\boldsymbol{F}} \sim \mathcal{N}\left(\boldsymbol{\mu}^d, \sigma^2 \boldsymbol{I}\right), \quad \text{where } \boldsymbol{\mu}^d = \mathrm{CompDec}\left(\boldsymbol{S}\right).
\tag{13}
$$

In the class prediction process, the final prediction is sampled from a Categorical distribution. The process is $y \sim \mathrm{Cat}(\boldsymbol{\pi})$ where

$$\pi_i = \frac{\exp\left(-\left\|\boldsymbol{S} - \sum_{n=1}^{N} \boldsymbol{T}_{i,n}/N\right\|_2^2\right)}{\sum_{l \in \mathcal{C}} \exp\left(-\left\|\boldsymbol{S} - \sum_{n=1}^{N} \boldsymbol{T}_{l,n}/N\right\|_2^2\right)}. \tag{14}$$

### A.3 DETAILS OF THE VARIATIONAL DISTRIBUTION

The variational distribution shares input and template encoding processes similar to the conditional generative process. The templates are encoded via $q(\boldsymbol{T}|\mathcal{T}, \boldsymbol{\epsilon}) = \prod_{i \in \mathcal{C}} \prod_{n=1}^{N} q(\boldsymbol{T}_{i,n}|\mathcal{T}_{i,n}, \boldsymbol{\epsilon})$, and the compositional latent components are extracted by

$$\tilde{\boldsymbol{S}} \sim \mathcal{N}\left(\tilde{\boldsymbol{\mu}}^s, \sigma^2 \boldsymbol{I}\right), \quad \text{where } \tilde{\boldsymbol{\mu}}^s = \mathrm{LCE}\left(\boldsymbol{X}\right),$$
$$\tilde{\boldsymbol{T}}_{i,n} \sim \mathcal{N}\left(\tilde{\boldsymbol{\mu}}_{i,n}^t, \sigma^2 \boldsymbol{I}\right), \quad \text{where } \tilde{\boldsymbol{\mu}}_{i,n}^t = \mathrm{LCE}\left(\mathcal{T}_{i,n}\right), \quad i \in \mathcal{C}, \quad n = 1, \cdots, N. \tag{15}$$

### A.4 MONTE CARLO ESTIMATOR OF THE ELBO

Using the detailed definition of the conditional generative process and variational distribution, the terms in the ELBO can be estimated by a Monte Carlo estimator as follows.

$$\mathcal{L}_{\mathrm{recon}} = \mathbb{E}_{q(\boldsymbol{S}|\boldsymbol{X},\boldsymbol{\epsilon})}\left[\log p(\boldsymbol{F}|\boldsymbol{S})\right] \approx -\frac{1}{2\sigma^2}\left\|\boldsymbol{F} - \mathrm{CompDec}(\tilde{\boldsymbol{S}})\right\|_2^2 + C(\sigma),$$

$$\mathcal{L}_{\mathrm{pred}} = \mathbb{E}_{q(\boldsymbol{S},\boldsymbol{T}|\boldsymbol{X},\mathcal{T},\boldsymbol{\epsilon})}\left[\log p(y|\boldsymbol{S},\boldsymbol{T})\right] \approx \log \frac{\exp\left(-\left\|\tilde{\boldsymbol{S}} - \sum_{n=1}^{N} \tilde{\boldsymbol{T}}_{y,n}/N\right\|_2^2\right)}{\sum_{l \in \mathcal{C}} \exp\left(-\left\|\tilde{\boldsymbol{S}} - \sum_{n=1}^{N} \tilde{\boldsymbol{T}}_{l,n}/N\right\|_2^2\right)}, \tag{16}$$

where $C(\sigma)$ is a constant related to the standard deviation $\sigma$, $\tilde{\boldsymbol{S}}$ and $\tilde{\boldsymbol{T}}$ are compositional latent components sampled from the variational distribution through Equation 15. Since the conditional generative process and the variational distribution share the same backbone and slot attention module in image encoding and template encoding, we have $p(\boldsymbol{S}|\boldsymbol{X}, \boldsymbol{\epsilon}) = q(\boldsymbol{S}|\boldsymbol{X}, \boldsymbol{\epsilon})$ and $p(\boldsymbol{T}_{i,n}|\mathcal{T}_{i,n}, \boldsymbol{\epsilon}) = q(\boldsymbol{T}_{i,n}|\mathcal{T}_{i,n}, \boldsymbol{\epsilon})$ if the same input and templates are given. Then, the two regularizers are

$$\mathcal{R}_{\mathrm{input}} = \mathrm{KL}\left(q(\boldsymbol{S}|\boldsymbol{X}, \boldsymbol{\epsilon}) \,\|\, p(\boldsymbol{S}|\boldsymbol{X}, \boldsymbol{\epsilon})\right) = 0,$$

$$\mathcal{R}_{\mathrm{temp}} = \mathrm{KL}\left(q(\boldsymbol{T}|\mathcal{T}, \boldsymbol{\epsilon}) \,\|\, p(\boldsymbol{T}|\mathcal{T}, \boldsymbol{\epsilon})\right) = \sum_{i \in \mathcal{C}} \sum_{n=1}^{N} \mathrm{KL}\left(q(\boldsymbol{T}_{i,n}|\mathcal{T}_{i,n}, \boldsymbol{\epsilon}) \,\|\, p(\boldsymbol{T}_{i,n}|\mathcal{T}_{i,n}, \boldsymbol{\epsilon})\right) = 0. \tag{17}$$

The Monte Carlo estimation of the ELBO is given by

$$\mathrm{ELBO}_{MC} = -\frac{1}{2\sigma^2}\left\|\boldsymbol{F} - \mathrm{CompDec}(\tilde{\boldsymbol{S}})\right\|_2^2 + \log \frac{\exp\left(-\left\|\tilde{\boldsymbol{S}} - \frac{\sum_{n=1}^{N} \tilde{\boldsymbol{T}}_{c,n}}{N}\right\|_2^2\right)}{\sum_{l \in \mathcal{C}} \exp\left(-\left\|\tilde{\boldsymbol{S}} - \frac{\sum_{n=1}^{N} \tilde{\boldsymbol{T}}_{l,n}}{N}\right\|_2^2\right)}. \tag{18}$$

We introduce a hyperparameter $\lambda$ to balance the importance of different terms:

$$\mathcal{L} = \left\|\boldsymbol{F} - \mathrm{CompDec}(\tilde{\boldsymbol{S}})\right\|_2^2 - \lambda \log \frac{\exp\left(-\left\|\tilde{\boldsymbol{S}} - \sum_{n=1}^{N} \tilde{\boldsymbol{T}}_{c,n}/N\right\|_2^2\right)}{\sum_{l \in \mathcal{C}} \exp\left(-\left\|\tilde{\boldsymbol{S}} - \sum_{n=1}^{N} \tilde{\boldsymbol{T}}_{l,n}/N\right\|_2^2\right)}. \tag{19}$$

## B DATASETS

Figure 5 visualizes the data used in the experiments. The Printed dataset is constructed on the basis of different printed font files. The handwritten dataset HWDB includes handwritten samples from

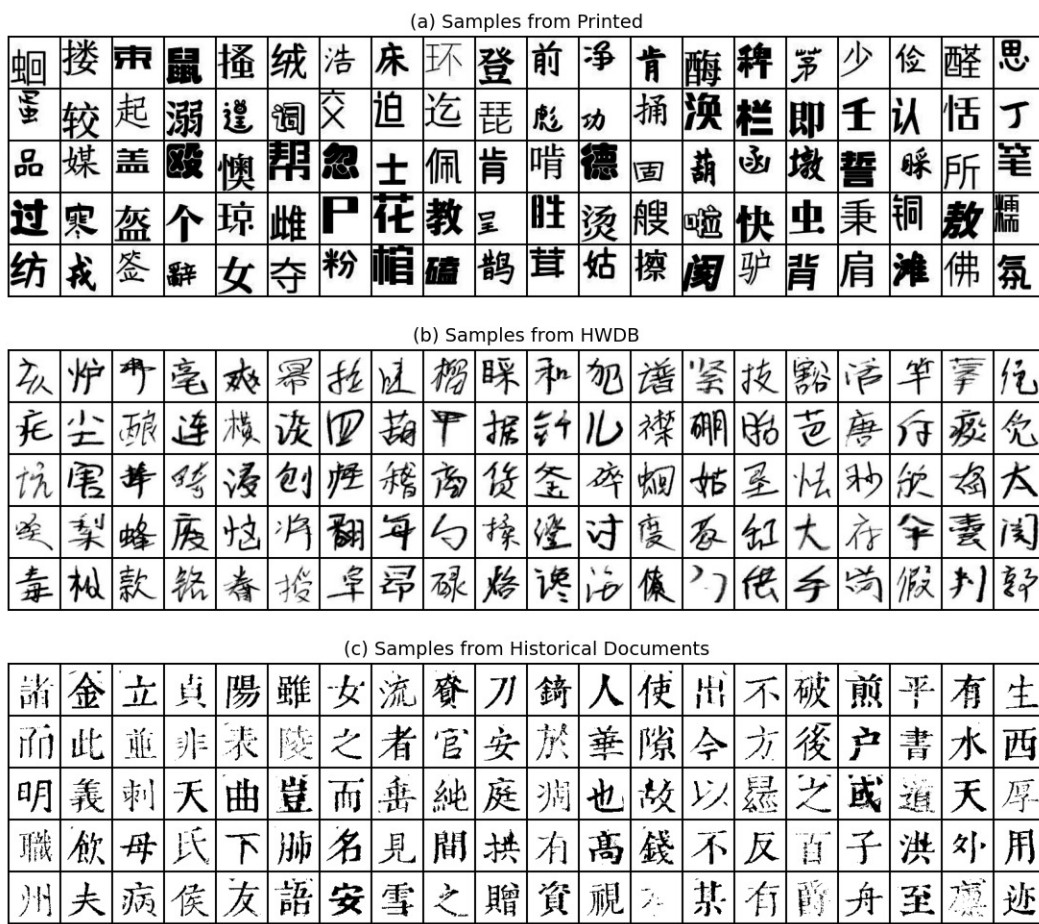

Figure 5: Visualization of the samples from (a) Printed, (b) HWDB and (c) Historical Documents.

various writers. The template set $\mathcal{T}$ is constructed by selecting $N = 10$ images for each character in the character set during the training and testing phases. The templates are generated from commonly used printed fonts to ensure that some rare characters also have templates. The character images in historical documents are collected from the web library[1]. In addition, all character images are allowed to used for free without any copyright concerns.

## C  DETAILS OF THE MODELS

The compared models follow the original experimental settings and architectures. We use open-source codes if the reimplemented methods have official codes (*e.g.*, CCR-CLIP and SD). CoLa is trained on the server with Intel(R) Xeon(R) Gold 6326 CPUs, 24GB NVIDIA GeForce RTX 4090 GPUs, 256GB RAM, and Ubuntu 20.04.6 LTS. CoLa is implemented with PyTorch (Paszke et al., 2019).

In the following, we will describe the architectures of learnable networks in CoLa and the hyperparameter selection of CoLa. The learnable networks include: (1) the CNN-based backbone; (2) the Spatial Broadcast Decoder (SBD) and the following linear layer to compose the components; (3) the two-layer CNN and prediction head of the teacher encoder. The details of the learnable networks are provided below.

- **The CNN-based backbone**:

---

[1]https://library.harvard.edu/policy-access-digital-reproductions-works-public-domain

- $5 \times 5$ Conv, stride 2, padding 2, 192, ReLU
- $5 \times 5$ Conv, stride 1, padding 2, 192, ReLU
- $5 \times 5$ Conv, stride 1, padding 2, 192, ReLU
- $5 \times 5$ Conv, stride 1, padding 2, 192
- Cartesian Positional Embedding, 192, LayerNorm
- Fully Connected, 192, ReLU
- Fully Connected, 192

We set the dimension of each component as 128, and the iteration step of updating component as 3. The components have the shape of $K \times 128$, initialized with the $\epsilon$ sampled from a learnable Gaussian $\mathcal{N}(\boldsymbol{\mu}_\epsilon, \boldsymbol{\sigma}_\epsilon^2)$.

- **SBD**:
    - Fully Connected, 192
    - Learnable 2D Positional Embedding, 192
    - Fully Connected, 1024, ReLU
    - Fully Connected, 1024, ReLU
    - Fully Connected, 1024, ReLU
    - Fully Connected, 1025
- **The composition linear layer after SBD**:
    - Fully Connected, 1024, without bias

The SBD output has the shape of $1025 \times 16 \times 16$, split along the channel dimension into the mask of $1 \times 16 \times 16$ and the component feature of $1024 \times 16 \times 16$. Here, $1024 \times 16 \times 16$ is the feature size of the teacher encoder.

- **The two-layer CNN of the teacher encoder**:
    - $3 \times 3$ Conv, stride 1, padding 1, 1024, ReLU
    - $3 \times 3$ Conv, stride 1, padding 1, 1024

The two-layer CNN converts the DINOv2 features of $768 \times 16 \times 16$ to the teacher features of $1024 \times 16 \times 16$. The teacher encoder is followed by a Transformer encoder block with a class token to predict the label of the input image. The DINOv2 encoder is frozen, and the two-layer CNN and prediction head are trained via a cross-entropy loss based on the training characters and the class labels. Finally, we use the output of the two-layer CNN as the teacher encoder, and the prediction head is not used in the following stages.

We train the teacher encoder using an Adam optimizer (Kingma & Ba, 2015), setting the learning rate to $3 \times 10^{-4}$ and the batch size to 8. Then we freeze the teacher encoder to train the remaining part of CoLa. We set the learning rate to $3 \times 10^{-4}$ and the batch size to 32. We first warm up CoLa by disabling the prediction term (i.e., setting $\lambda = 0$). After the training loss is stable, we enable the prediction term by setting the $\lambda = 0.01$ to train the model. The template images in the training character set are also used to train CoLa in the training process. The CNN-based backbone and the slot attention module are frozen when extracting components of templates to stop the gradient propagation to reduce the cost of computational resources.

# D  ADDITIONAL EXPERIMENTAL RESULTS

## D.1  ADDITIONAL RESULTS OF COMPONENT VISUALIZATION

To further display the compositional latent components learned by CoLa, Figure 6 provides additional visualization results on three different datasets: (a) Printed, (b) HWDB, and (c) Historical Document. We select input images from the test set of each dataset and visualize their compositional latent components (Cmp#1-#3). CoLa can decompose characters into meaningful structures in most cases, despite various character styles across the datasets. On images from historical documents (Figure 6c), CoLa still successfully extracts compositional latent components, even if some images are incomplete or partly ambiguous. These results demonstrate that CoLa can learn interpretable components across different styles of Chinese characters and handle low-quality images.

## D.2 INFLUENCE OF THE COMPONENT ORDER

This experiment investigates how the decomposition order $\epsilon$ influences the learning of compositional latent representations. Figure 7 compares the components learned in different ways of initialization: (a) using random initialization, where CoLa samples $\epsilon$ from the Gaussian distribution randomly for each example; (b) using fixed initialization, where a fixed order is used for all examples. For each initialization method, we visualize four groups of character images with their compositional latent components (Cmp#1-#3). In Figure 7a, the components learned with random initialization have similar decomposition, but the component order varies across examples. With a fixed order (In Figure 7b), CoLa assigns semantically or spatially similar components in the same order for one character. These results demonstrate that CoLa's decomposition process is controlled by the order $\epsilon$, which works by initializing the latent component with $\epsilon$ before input into the slot attention module.

## D.3 AVERAGE INFERENCE TIME

To evaluate the computational efficiency of the models, we estimate the per-batch inference time required for predicting the label of input images on a dataset with 3,755 classes of characters. We set the batch size to 32 and compute the average inference time across all batches in the test set during the evaluation phase. As shown in Table 4, CoLa demonstrates higher time efficiency compared to previous zero-shot Chinese character recognition (CCR) methods.

Table 4: Comparison of average inference time (AIT).

| Methods | Ours | DenseRAN | HDE | SD | CCR-CLIP |
|---------|------|----------|-----|-----|----------|
| AIT(ms) | **9** | 1666 | 29 | 567 | 14 |

## D.4 HYPERPARAMETER SELECTION

As shown in Table 5, we examine two hyperparameters that influence the performance of CoLa: the number of components $K$ and the number of template character images used for matching $N$.

**Number of Components.** On the HWDB and Printed datasets, CoLa achieves its best performance when the number of components $K = 3$. Our results indicate that continuously increasing $K$ does not lead to further performance gains. Instead, once the model surpasses an optimal level of decomposition, additional components may even degrade performance (e.g., $K > 3$). The excessively fine-grained partitioning may introduce redundant or noisy substructures, and expanding $K$ may in fact hinder the ability of CoLa to generalize. This observation is consistent with cognitive evidence in Chinese character recognition, which indicates that native readers tend to mentally decompose a character into a small number of interpretable structural units, rather than a large set of fine-grained details (Yeh & Li, 2002). In this sense, the component decomposition mechanism of CoLa implicitly aligns with the human processing strategy, where a small number of structural parts often suffice to capture the decomposition of Chinese characters.

**Number of Templates.** We further investigate how the number of template character images influences the performance of CoLa. Across both the HWDB and Printed datasets, expanding the template set consistently improves matching accuracy, suggesting that a richer reference pool enables the model to better capture intra-class variations and reduce ambiguity in zero-shot recognition. However, the benefit of adding more templates gradually diminishes as the number increases. In particular, performance gains become less when $N > 10$, indicating that the model has already saturated its ability to benefit from additional examples. Considering both the empirical results and the practical cost of maintaining larger template sets, we adopt $N = 10$ templates per class, which offers a trade-off between accuracy and efficiency.

## D.5 INFLUENCE OF TEMPLATE SAMPLING

Table 6 reports the standard deviation of recognition accuracy obtained using five independently constructed template sets. Each template set is generated from public printed fonts collected from the Internet. We manually filtered the candidate fonts and applied quantitative metrics to ensure

Table 5: **Recognition Accuracy (%) with different $K$ (#Comp) and $N$ (#Temp).** The table provides the performance of CoLa on both the handwritten dataset HWDB and the printed character datasets Printed.

| #Comp | HWDB (Character Zero-shot) | | | | | Printed (Character Zero-shot) | | | | |
|---|---|---|---|---|---|---|---|---|---|---|
| | 500 | 1000 | 1500 | 2000 | 2755 | 500 | 1000 | 1500 | 2000 | 2755 |
| $K=1$ | 41.19 | 39.30 | 46.28 | 47.97 | 53.82 | 78.08 | 83.24 | 87.03 | 88.05 | 85.40 |
| $K=2$ | 58.20 | 66.07 | 70.23 | 72.38 | 79.34 | 78.11 | 84.51 | 90.89 | 92.40 | 92.80 |
| $K=3$ | **68.59** | **76.58** | **79.16** | **81.16** | **82.71** | 78.10 | **85.38** | 90.32 | **93.26** | 92.70 |
| $K=4$ | 58.08 | 69.67 | 73.99 | 75.34 | 82.16 | 78.24 | 84.38 | 90.82 | 92.68 | **93.61** |
| $K=5$ | 55.44 | 68.85 | 71.86 | 74.32 | 81.09 | **78.26** | 84.24 | **90.94** | 92.36 | 93.29 |

| #Comp | HWDB (Radical Zero-shot) | | | | | Printed (Radical Zero-shot) | | | | |
|---|---|---|---|---|---|---|---|---|---|---|
| | 50 | 40 | 30 | 20 | 10 | 50 | 40 | 30 | 20 | 10 |
| $K=1$ | 39.58 | 45.77 | 55.26 | 52.00 | 49.44 | 79.79 | 80.82 | 82.79 | 87.57 | 89.22 |
| $K=2$ | 64.10 | 68.83 | 75.18 | 75.44 | 68.12 | 82.46 | 84.54 | 86.87 | 90.85 | 94.37 |
| $K=3$ | **70.40** | **74.80** | 77.01 | **80.64** | **75.78** | 82.23 | 84.48 | 82.20 | **92.12** | **94.81** |
| $K=4$ | 69.62 | 74.43 | **78.25** | 80.62 | 73.09 | **82.31** | **85.00** | **87.34** | 91.96 | 94.65 |
| $K=5$ | 68.77 | 73.01 | 77.43 | 79.23 | 62.00 | 82.22 | 84.88 | 86.79 | 91.34 | 94.39 |

| #Temp | HWDB (Character Zero-shot) | | | | | Printed (Character Zero-shot) | | | | |
|---|---|---|---|---|---|---|---|---|---|---|
| | 500 | 1000 | 1500 | 2000 | 2755 | 500 | 1000 | 1500 | 2000 | 2755 |
| $N=1$ | 41.05 | 51.56 | 56.23 | 61.32 | 65.82 | 60.39 | 69.52 | 81.50 | 84.13 | 85.73 |
| $N=3$ | 41.99 | 52.58 | 59.23 | 60.65 | 70.44 | 64.14 | 70.66 | 82.26 | 86.49 | 87.71 |
| $N=5$ | 42.46 | 54.33 | 60.08 | 59.89 | 71.08 | 69.57 | 77.55 | 86.57 | 89.09 | 90.86 |
| $N=10$ | **68.59** | 76.58 | 79.16 | **81.16** | 82.71 | 78.10 | 85.38 | 90.32 | 93.26 | 92.70 |
| $N=20$ | 66.92 | **76.67** | **79.94** | 81.14 | **84.11** | **80.31** | **86.25** | **91.98** | **93.69** | **94.36** |

| #Temp | HWDB (Radical Zero-shot) | | | | | Printed (Radical Zero-shot) | | | | |
|---|---|---|---|---|---|---|---|---|---|---|
| | 50 | 40 | 30 | 20 | 10 | 50 | 40 | 30 | 20 | 10 |
| $N=1$ | 48.12 | 53.88 | 58.93 | 61.04 | 63.45 | 64.26 | 69.85 | 79.45 | 83.88 | 87.00 |
| $N=3$ | 50.36 | 60.19 | 63.38 | 67.01 | 67.70 | 71.25 | 72.26 | 81.61 | 85.60 | 89.96 |
| $N=5$ | 53.53 | 60.70 | 63.87 | 66.91 | 61.89 | 75.52 | 77.75 | 83.55 | 88.41 | 92.45 |
| $N=10$ | 70.40 | 74.80 | 77.01 | 80.64 | 75.78 | 82.23 | 84.48 | 82.20 | 92.12 | 94.81 |
| $N=20$ | **74.27** | **78.07** | **81.19** | **82.70** | **83.42** | **83.43** | **87.11** | **91.63** | **93.07** | **95.36** |

Table 6: **Standard deviation of recognition accuracy under different template sets.** We evaluate CoLa on HWDB and Printed using five independently constructed template sets, each generated from ten public printed fonts ($N = 10$). Each cell reports the standard deviation of recognition accuracy across the five template sets.

| Datasets | Character Zero-shot | | | | | Radical Zero-shot | | | | |
|---|---|---|---|---|---|---|---|---|---|---|
| | 500 | 1000 | 1500 | 2000 | 2755 | 50 | 40 | 30 | 20 | 10 |
| HWDB | 2.63 | 2.16 | 1.63 | 1.98 | 0.75 | 1.54 | 1.30 | 1.04 | 1.15 | 1.68 |
| Printed | 1.56 | 1.34 | 0.74 | 0.47 | 0.50 | 1.58 | 1.20 | 0.74 | 0.61 | 0.32 |

usability. In particular, we manually examined some character images rendered from each font file and computed statistics such as the number and spatial distribution of black pixels in each image. This procedure ensures that the remaining template fonts produce visually correct character images.

The results in Table 6 show that the variance across different template sets is influenced by the size of the training character set. The overall influence is limited, especially when the training character set is large. When only a small number of classes are observed during training, e.g., 500 characters, the model's recognition accuracy exhibits higher sensitivity to template selection. As the training set expands, these variations progressively diminish. When the model has been trained on all 2,755 characters, the effect of different template sets is attenuated, with deviations reduced to 0.75% on HWDB and 0.50% on Printed. A similar trend emerges in the radical zero-shot scenario,

Table 7: **Performance of CoLa on the fine-grained image categorization dataset Stanford Dogs.** We evaluate the generalization ability of CoLa on the vision task beyond Chinese character recognition. Approximately 10 out of 120 classes are held out as unseen test classes. The entire dataset contains around 20,000 images of different dog breeds.

| Models | Random Baseline | CLIP | DINOv2 | Slot Attention | CoLa |
|---|---|---|---|---|---|
| Accuracy | 0.83 | 0.01 | 7.72 | 7.34 | **22.26** |

Table 8: **Quantitative results on cross-script recognition.** "Conf." denotes the cross-script settings, where "A → B" means that a model trained on A is evaluated on B. We report multiple commonly used retrieval metrics: Recall@K, Prec@K, F1@K, and MRR.

| Models | Conf. | Recall@1 | Recall@5 | Prec@1 | Prec@5 | F1@1 | F1@5 | MRR |
|---|---|---|---|---|---|---|---|---|
| CLIP | | 0.67 | 3.36 | 0.67 | 0.67 | 0.67 | 1.12 | 10.45 |
| DINOv2 | | 64.07 | 90.65 | 64.07 | 53.49 | 64.07 | 63.84 | 75.79 |
| DINOv2 Ft. | zh → ja | 79.01 | 96.47 | 79.01 | 70.80 | 79.01 | 78.55 | 86.64 |
| Slot Attention | | 76.67 | 95.52 | 76.67 | 71.42 | 76.67 | 78.42 | 84.85 |
| CoLa | | **83.03** | **97.95** | **83.03** | **77.77** | **83.03** | **83.99** | **89.62** |
| CLIP | | 0.07 | 0.33 | 0.07 | 0.07 | 0.07 | 0.11 | 9.22 |
| DINOv2 | | 3.98 | 10.17 | 3.98 | 2.27 | 3.98 | 3.66 | 14.56 |
| DINOv2 Ft. | zh → ko | 11.97 | 30.61 | 11.97 | 8.56 | 11.97 | 12.91 | 25.34 |
| Slot Attention | | 13.42 | 30.60 | 13.42 | 8.05 | 13.42 | 12.36 | 26.19 |
| CoLa | | **15.03** | **39.58** | **15.03** | **11.71** | **15.03** | **17.40** | **29.89** |
| CLIP | | 0.01 | 0.06 | 0.01 | 0.01 | 0.01 | 0.02 | 9.12 |
| DINOv2 | | 49.78 | 65.88 | 49.78 | 25.54 | 49.78 | 34.84 | 59.26 |
| DINOv2 Ft. | zh → obc | 53.80 | 69.68 | 53.80 | 28.60 | 53.80 | 38.32 | 62.83 |
| Slot Attention | | 45.97 | 59.30 | 45.97 | 21.83 | 45.97 | 30.19 | 54.87 |
| CoLa | | **61.97** | **74.84** | **61.97** | **33.67** | **61.97** | **43.88** | **69.39** |

where the standard deviation gradually drops as the number of recognizable radicals increases. This behavior indicates that once the model has encountered sufficient structural diversity during training, it becomes increasingly robust to the style difference in templates.

### D.6 GENERALIZE TO OTHER TASKS

We evaluate the generalization ability of CoLa beyond the CCR task by applying it to the Stanford Dogs dataset (Khosla et al., 2011). Since the compared CCR method cannot be directly applied to the dataset, we compare CoLa with three representative models: CLIP, DINOv2, and Slot Attention. Stanford Dogs is a fine-grained categorization dataset with small intra-class variations and strong visual similarity across breeds. We hold out 10 out of 120 types of dogs as unseen test classes, ensuring that no images from these categories appear during training for zero-shot evaluation.

The results in Table 7 show that CoLa achieves the best performance, with CLIP's performance even falling below the random baseline. This contrast highlights an important limitation of CLIP. The CLIP features tend to emphasize high-level semantic concepts (i.e., recognizing that all samples are "dogs"), while ignoring the fine-grained information to distinguish different dog categories. The compositional latent components of CoLa can capture variations between dogs for zero-shot classification on this fine-grained dataset.

### D.7 QUANTITATIVE RESULTS ON CROSS-SCRIPT EVALUATION

To assess the model performance on the Japanese, Korean, and OBC datasets, we employ several metrics that are commonly used in zero-shot retrieval tasks. Recall@K measures whether at least one correct class appears within the top-K matched templates, and is therefore an appropriate indicator of match success in the zero-shot setting. Prec@K quantifies the proportion of correctly matched templates among the top-K results, which is informative when multiple examples exist for each class. To jointly capture both aspects, we further report F1@K, which is the harmonic mean of Prec@K

Table 9: **Performance of cross-style Chinese character recognition.** For the configuration HWDB → Printed, CoLa is trained on handwritten characters and evaluated on printed characters.

| Configurations | Character Zero-shot | | | | | Radical Zero-shot | | | | |
|---|---|---|---|---|---|---|---|---|---|---|
| | 500 | 1000 | 1500 | 2000 | 2755 | 50 | 40 | 30 | 20 | 10 |
| HWDB → Printed | 66.88 | 72.44 | 71.31 | 71.78 | 76.65 | 73.66 | 71.53 | 73.80 | 77.32 | 82.02 |
| Printed → HWDB | 11.69 | 15.85 | 24.75 | 29.59 | 27.99 | 14.09 | 15.93 | 23.06 | 30.59 | 29.35 |

and Recall@K. In addition, we employ the Mean Reciprocal Rank (MRR) to evaluate how early the correct class appears in the ranked candidates. MRR is useful when distinguishing between visually similar characters, as it rewards models that place the correct match at higher ranks. Table 8 presents the quantitative results of different cross-script character recognition configurations. For example, zh → ja indicates that the model is trained on Chinese characters and evaluated on Japanese characters. Across the configurations, we observe that CoLa outperforms other baselines, demonstrating its generalization ability in different writing systems.

The DINOv2 backbone significantly outperforms CLIP, indicating that features learned by DINOv2 are more suitable for character matching or recognition. CLIP's visual encoder is trained with large-scale contrastive supervision, which may encourage global cross-modal alignment rather than fine-grained discrimination. DINOv2 is optimized on massive image corpora, and therefore learns mid-level and local structural patterns that better capture geometric and morphological variations among characters. These signals are crucial for distinguishing characters, which explains the gap between DINOv2 and CLIP in different metrics. Fine-tuning DINOv2 with a character classification objective further improves performance. As shown in Table 8 (i.e., DINOv2 Ft.), adapting the encoder to the specific domain yields more discriminative features. The supervised signal refines the feature space such that characters with similar shapes and structures cluster more tightly. This domain adaptation benefits cross-script generalization, as fine-tuned DINOv2 surpasses the frozen backbone consistently across the metrics. CoLa outperforms Slot Attention despite both methods learning component representations. Slot Attention learns components by primarily minimizing reconstruction loss, while CoLa integrates compositional latent components with template-based matching and feature-level reconstruction. As a result, CoLa achieves improvements in Recall@K, Prec@K, F1@K, and MRR, especially in the setting zh → obc, where the differences between scripts are substantial.

## D.8 CROSS-STYLE EVALUATION

Table 9 reports the quantitative results of cross-style Chinese character recognition. The configurations indicate different training-testing settings, where a model trained on one style (e.g., handwritten) is directly evaluated on another (e.g., printed). We observe that CoLa still maintains recognition ability under cross-style transfer. For example, transferred from HWDB to Printed, CoLa achieves 66.88% accuracy on the 500-character split and increases to 76.65% for the 2755-character one. This demonstrates that the learned compositional latent components can generalize across different visual styles without requiring re-training or adaptation.

The results also suggest that cross-style performance is related to the diversity and complexity of the training data. Handwritten characters exhibit larger structural variation and irregular stroke distribution, forcing the model to learn more flexible relational or structural features. Printed characters are visually homogeneous and contain less intra-class variability. When trained on handwritten data (HWDB), CoLa is encouraged to discover structures that remain informative under style variations, which can generalize to printed fonts. In contrast, the performance of CoLa drops more significantly on HWDB → Printed. This asymmetry reflects that models trained on more complex and diverse sources tend to transfer better to simpler or more regular targets, but not vice versa. Printed glyphs may lack the variability necessary to teach the model how to handle large deformations, stroke variance, or handwriting artifacts.

Table 10: **Performance of CoLa trained with different teacher models.** "Ft." indicates that the pretrained backbone is further fine-tuned using the corresponding training data.

| Teachers | Ft. | HWDB (Radical Zero-shot) | | | | |
|---|---|---|---|---|---|---|
| | | 50 | 40 | 30 | 20 | 10 |
| CLIP$_{\text{ViT-B/16}}$ | ✓ | 49.96 | 59.36 | 60.60 | 64.57 | 66.60 |
| DINOv2$_{\text{ViT-B/16}}$ | | 66.33 | 72.15 | 74.09 | 76.81 | **78.29** |
| DINOv2$_{\text{ViT-B/16}}$ (CoLa) | ✓ | **70.40** | **74.80** | **77.01** | **80.64** | 75.78 |

### D.9 INFLUENCE OF TEACHER MODELS

Table 10 summarizes the performance of CoLa when trained with different teacher encoders, where DINOv2 outperforms CLIP under most configurations. CLIP may focus more on high-level semantic alignment and tend to downplay local structural cues. Since CoLa relies on component-level comparison, representations that retain detailed visual signals offer more effective supervision, which explains the consistent performance advantage of DINOv2 over CLIP. Fine-tuning the pre-trained encoder with the character classification objective improves performance. When DINOv2 is fine-tuned on character images (DINOv2 ViT-B/16 + Ft.), the model gains additional domain-specific priors. Although the fine-tuning task does not explicitly leverage compositional information of characters, it still organizes the feature space such that similar characters are pulled closer while dissimilar ones are pushed apart.

### D.10 ADDITIONAL VISUALIZATION RESULTS OF CHARACTER DECOMPOSITION

Figure 8 presents visualizations of CoLa on different Chinese character structures, e.g., left–right, up–down, and various surrounding structures. For two-component structures such as Left–Right (a) and Up–Down (c), CoLa produces clear component separation, often aligning its slots with radical-level subregions. The two-part layouts are commonly seen in Chinese characters, allowing the encoder to form stable component decomposition. In three-component layouts, such as Left–Middle–Right (b) and Up–Middle–Down (d), CoLa frequently produces two slots instead of three. This behavior reflects a preference toward grouping spatially adjacent radicals into a single perceptual unit. In surrounding structures (e–j), component decomposition becomes more challenging. CoLa produces more accurate decomposition results in Bottom-Left Surrounded and Top-Left Surrounded layouts, while it tends to treat the entire character as a single component in the Fully-Wrapping structure. The visualization results demonstrate that CoLa learns to decompose components that reflect radical-level semantics and spatial roles.

## E   LLM USAGE STATEMENT

We use Large Language Models (LLMs) as auxiliary tools during the preparation of this paper. The usage is limited to correcting grammatical issues, improving readability, and polishing the presentation. LLMs do not contribute to the generation of research ideas, experimental design, data analysis, or theoretical development. All scientific content and claims are produced by the authors.

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

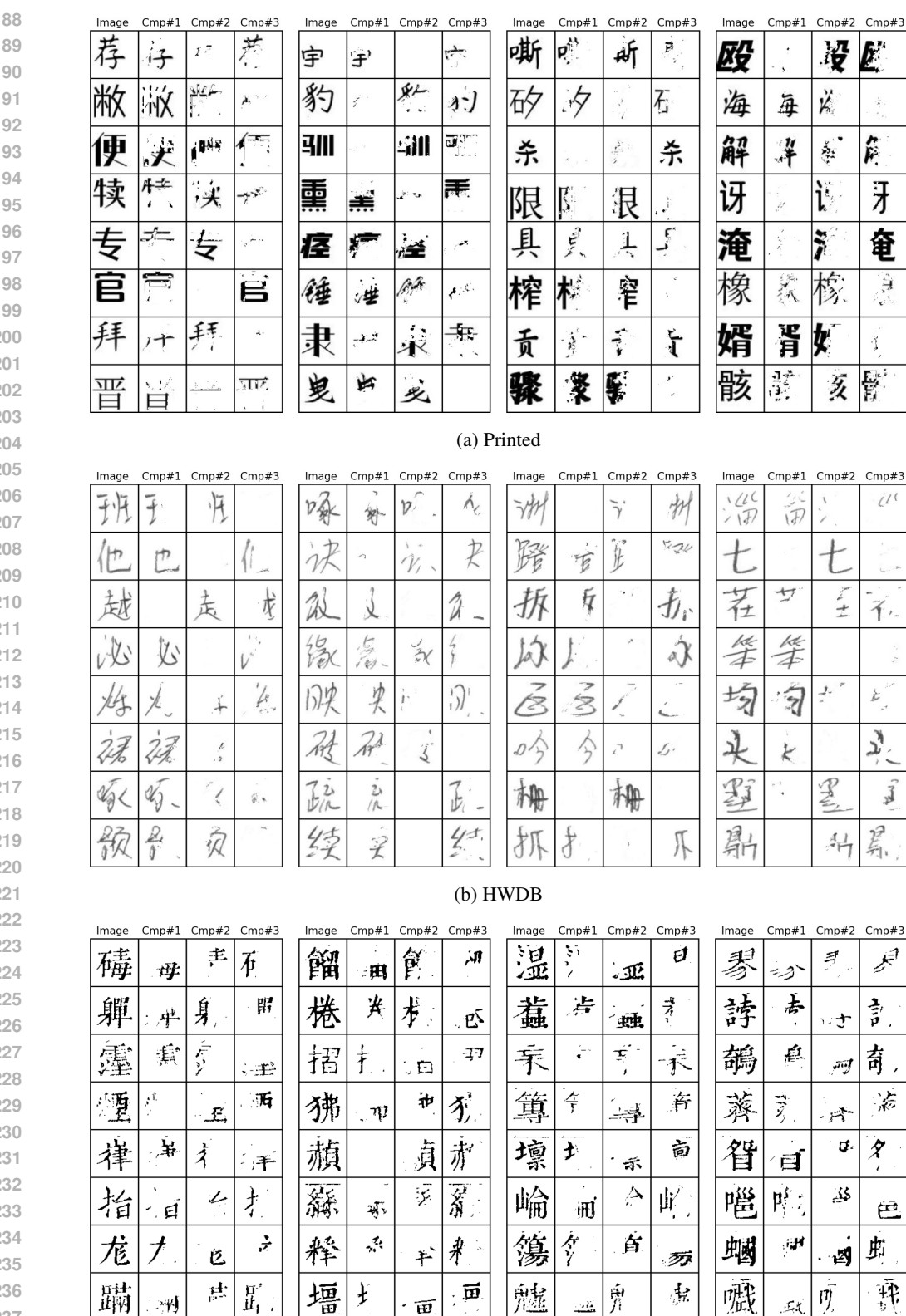

(a) Printed

(b) HWDB

(c) Historical Document

Figure 6: Visualization of the compositional latent components on the different datasets

(a) Random initialization

(b) Fix initialization

Figure 7: **The compositional latent components learned with different orders.** (a) Random initialization. The components in each panel are learned with randomly sampled $\epsilon$. (b) Fix initialization. The components in each panel are learned using a fixed $\epsilon$.

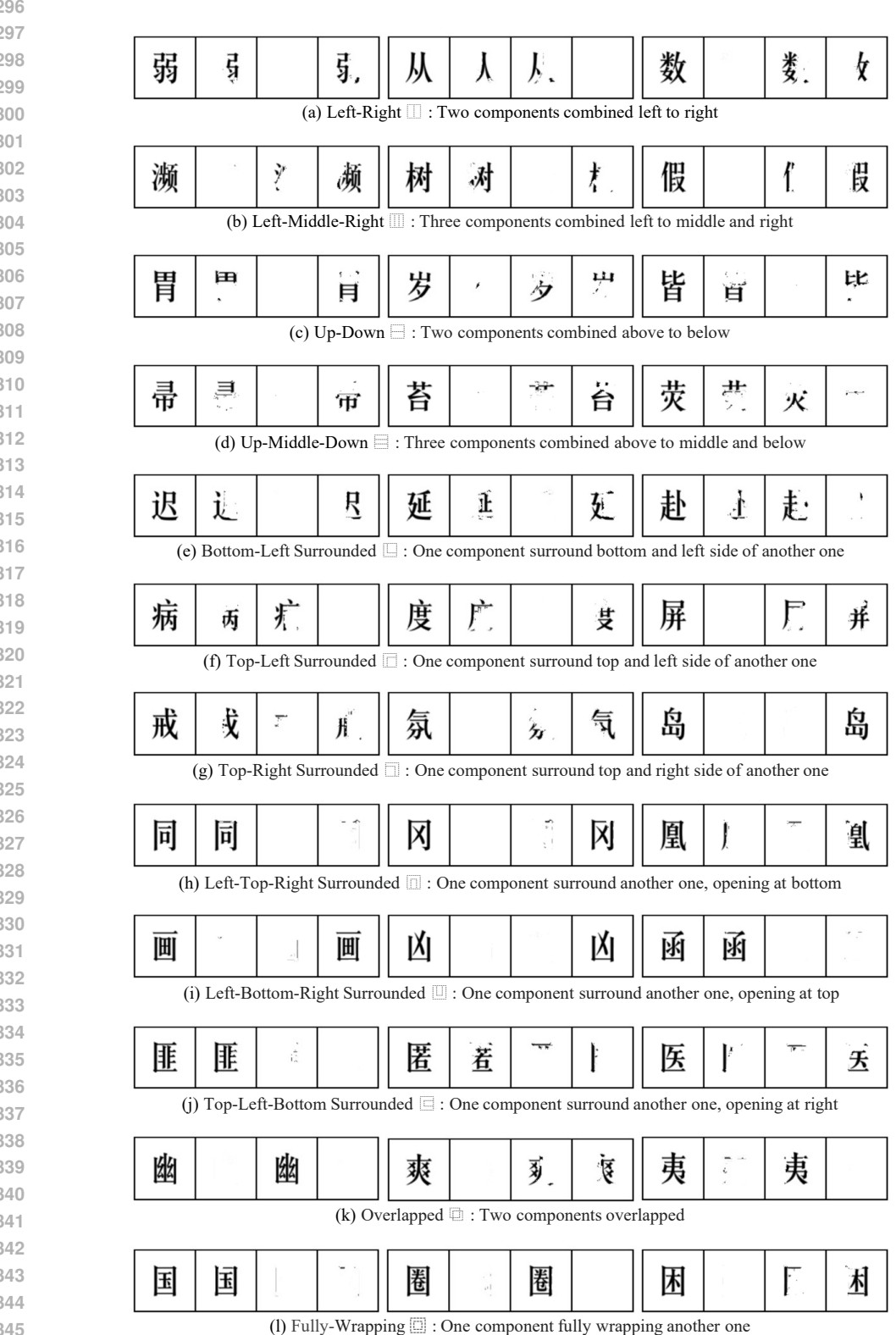

Figure 8: **Visualization of component decomposition.** The figures (a)-(l) show decomposition results of twelve character structures, respectively.

