# OpenReview forum: "Chinese Character Decomposition with Compositional Latent Components"
_ICLR.cc/2026/Conference — Submitted to ICLR 2026_

### Official Review · Reviewer_kKYY · 2025-10-28

**Soundness:** 3
**Presentation:** 2
**Contribution:** 2
**Rating:** 4
**Confidence:** 4

**Summary:**

The paper proposes CoLa, a compositional latent–component model for Chinese character recognition that eliminates manual part annotations. An encoder decomposes an input glyph into K permutation-invariant “slots,” a frozen teacher guides the slots via feature reconstruction, and classification is performed by measuring distances between the input’s slot representation and class “prototypes” computed from a small set of font-rendered templates per character. This design aims to support zero-shot recognition of unseen characters and rare components. Experiments report notable gains on zero-shot splits (including a constructed “component zero-shot” setting) and on harder domains such as historical documents; ablations indicate both the teacher-guided reconstruction and the prototype classifier contribute materially. Visualizations hint at cross-script transferability.

**Strengths:**

Pros:
1.	Clear, coherent pipeline (slots + teacher alignment + prototype matching).
2.	Removes reliance on handcrafted part labels; offers intuitive visualizations.
3.	Broad evaluations with ablations; strong reported gains on defined splits.

**Weaknesses:**

Cons:
1.	Motivation not substantiated with real-world evidence; zero-shot largely constructed.
2.	Heavy dependence on font-rendered templates and a closed label set; no open-set handling.
3.	Limited analysis of style/domain shift and cross-script transfer (mostly qualitative).
4.	Insufficient transparency: exact fonts, template policy, hyperparameter/cost sensitivity, and teacher dependence not systematically studied.

**Questions:**

Zero-shot claims rest critically on the availability of font templates and a closed candidate set, leaving open questions about domain/style gaps (fonts vs handwriting/archives), open-set rejection, and robustness. Important engineering details (font sources, template sampling, cost/speed, sensitivity to K and template count) are insufficiently reported, limiting reproducibility and practical credibility.

---

> ### Author Response · Authors · 2025-12-02
> **Response to Reviewer kKYY (Part 1)**
>
> We sincerely thank the reviewer for the constructive comments. We address the concerns as follows.
>
> **Q1. Motivation not substantiated with real-world evidence; zero-shot largely constructed.**
>
> We appreciate the reviewer’s concern about our motivation and zero-shot setup.
>
> The core topic of our work is learning compositional components from Chinese characters. In real-world scenarios, the supervised labels are sometimes absent, such as rare characters in historical documents, variant forms, or ancient scripts like oracle bone inscriptions that remain partially undeciphered. Compositional decomposition will help recognition in these cases by transferring the learned character decomposition scheme to unseen characters.
>
> Zero-shot CCR is a downstream task that demonstrates whether CoLa has learned reusable latent components. For example, in oracle bone inscriptions (OBC), approximately 4,500 distinct character classes have been discovered, but only about 2,200 have been reliably deciphered, meaning that more than half of the character inventory has no stable annotation. In this case, the zero-shot recognition ability is critical for handling characters that cannot be annotated or interpreted in advance.
>
>
>
> **Q2. Dependence on templates and closed candidate set**
>
> > Zero-shot claims rest critically on the availability of font templates and a closed candidate set
> >
> > Heavy dependence on font-rendered templates and a closed label set; no open-set handling.
>
>
>
> We thank the reviewer for this comment. The font-rendered templates are only used as visual references and can be substituted by any collection, such as historical manuscripts, printed fonts, or even user-provided samples. Considering the difficulty of collecting templates, the template set is generated from public and easy-to-fetch printed fonts from the Internet. Moreover, the overall influence of using different template sets is limited in our experiment. CoLa also supports open-set scenarios. New characters can be added by simply providing template images of the new class, without retraining or updating model parameters.

---

> > ### Author Response · Authors · 2025-12-02
> > **Response to Reviewer kKYY (Part 2)**
> >
> > **Q3. Style/domain shift and cross-script transfer**
> >
> > > leaving open questions about domain/style gaps (fonts vs handwriting/archives)
> > >
> > > Limited analysis of style/domain shift and cross-script transfer (mostly qualitative).
> >
> >
> >
> > **Style/Domain Shift Results**
> >
> > We **report the quantitative results of cross-style Chinese character recognition** in the following two tables. The configurations indicate different training-testing settings, where a model trained on one style (e.g., handwritten) is directly evaluated on another (e.g., printed). We observe that CoLa still maintains recognition ability under cross-style transfer. For example, transferred from HWDB to Printed, CoLa achieves 66.88% accuracy on the 500-character split and increases to 76.65% for the 2755-character one. This demonstrates that the learned compositional latent components can generalize across different visual styles without requiring re-training or adaptation. When trained on handwritten data (HWDB), CoLa is encouraged to discover structures that remain informative under style variations, which can generalize to printed fonts. In contrast, the performance of CoLa drops more significantly on HWDB
> >
> > \rightarrow$ Printed.
> >
> > | Character Zero-shot Settings |  500  | 1000  | 1500  | 2000  | 2755  |
> > | ---------------------------- | :---: | :---: | :---: | :---: | :---: |
> > | HWDB → Printed               | 66.88 | 72.44 | 71.31 | 71.78 | 76.65 |
> > | Printed → HWDB               | 11.69 | 15.85 | 24.75 | 29.59 | 27.99 |
> >
> > | Radical Zero-shot Settings |  50   |  40   |  30   |  20   |  10   |
> > | -------------------------- | :---: | :---: | :---: | :---: | :---: |
> > | HWDB → Printed             | 73.66 | 71.53 | 73.80 | 77.32 | 82.02 |
> > | Printed → HWDB             | 14.09 | 15.93 | 23.06 | 30.59 | 29.35 |
> >
> >
> >
> > **Cross-script Transfer Results**
> >
> > We **assess the cross-script transfer performance on the Japanese, Korean, and OBC datasets** in the following tables, using several metrics that are commonly used in zero-shot retrieval tasks. Recall@K measures whether at least one correct class appears within the top-K matched templates, and is therefore an appropriate indicator of match success in the zero-shot setting. Prec@K quantifies the proportion of correctly matched templates among the top-K results, which is informative when multiple examples exist for each class. To jointly capture both aspects, we further report F1@K, which is the harmonic mean of Prec@K and Recall@K. In addition, we employ the Mean Reciprocal Rank (MRR) to evaluate how early the correct class appears in the ranked candidates. Across the configurations, we observe that CoLa outperforms other baselines, demonstrating its generalization ability in different writing systems.
> >
> > | Models (Japanese) | Recall@1  | Recall@5  |  Prec@1   |  Prec@5   |   F1@1    |   F1@5    |    MRR    |
> > | :---------------- | :-------: | :-------: | :-------: | :-------: | :-------: | :-------: | :-------: |
> > | CLIP              |   0.67    |   3.36    |   0.67    |   0.67    |   0.67    |   1.12    |   10.45   |
> > | DINOv2            |   64.07   |   90.65   |   64.07   |   53.49   |   64.07   |   63.84   |   75.79   |
> > | DINOv2 Ft.        |   79.01   |   96.47   |   79.01   |   70.80   |   79.01   |   78.55   |   86.64   |
> > | Slot Attention    |   76.67   |   95.52   |   76.67   |   71.42   |   76.67   |   78.42   |   84.85   |
> > | **CoLa**          | **83.03** | **97.95** | **83.03** | **77.77** | **83.03** | **83.99** | **89.62** |
> >
> > | Models (Korean) | Recall@1  | Recall@5  |  Prec@1   |  Prec@5   |   F1@1    |   F1@5    |    MRR    |
> > | :-------------- | :-------: | :-------: | :-------: | :-------: | :-------: | :-------: | :-------: |
> > | CLIP            |   0.07    |   0.33    |   0.07    |   0.07    |   0.07    |   0.11    |   9.22    |
> > | DINOv2          |   3.98    |   10.17   |   3.98    |   2.27    |   3.98    |   3.66    |   14.56   |
> > | DINOv2 Ft.      |   11.97   |   30.61   |   11.97   |   8.56    |   11.97   |   12.91   |   25.34   |
> > | Slot Attention  |   13.42   |   30.60   |   13.42   |   8.05    |   13.42   |   12.36   |   26.19   |
> > | **CoLa**        | **15.03** | **39.58** | **15.03** | **11.71** | **15.03** | **17.40** | **29.89** |
> >
> > | Models (OBC)   | Recall@1  | Recall@5  |  Prec@1   |  Prec@5   |   F1@1    |   F1@5    |    MRR    |
> > | :------------- | :-------: | :-------: | :-------: | :-------: | :-------: | :-------: | :-------: |
> > | CLIP           |   0.01    |   0.06    |   0.01    |   0.01    |   0.01    |   0.02    |   9.12    |
> > | DINOv2         |   49.78   |   65.88   |   49.78   |   25.54   |   49.78   |   34.84   |   59.26   |
> > | DINOv2 Ft.     |   53.80   |   69.68   |   53.80   |   28.60   |   53.80   |   38.32   |   62.83   |
> > | Slot Attention |   45.97   |   59.30   |   45.97   |   21.83   |   45.97   |   30.19   |   54.87   |
> > | **CoLa**       | **61.97** | **74.84** | **61.97** | **33.67** | **61.97** | **43.88** | **69.39** |

---

> > > ### Author Response · Authors · 2025-12-02
> > > **Response to Reviewer kKYY (Part 3)**
> > >
> > > **Q4. Model details**
> > >
> > > > Insufficient transparency: exact fonts, template policy, hyperparameter/cost sensitivity, and teacher dependence not systematically studied.
> > > >
> > > > Important engineering details (font sources, template sampling, cost/speed, sensitivity to K and template count) are insufficiently reported, limiting reproducibility and practical credibility.
> > >
> > >
> > >
> > > **Details of template sampling**
> > >
> > > The following table reports the standard deviation of recognition accuracy obtained using five independently constructed template sets. The results show that the variance across different template sets is influenced by the size of the training character set. **The overall influence is limited, especially when the training character set is large.** When only a small number of classes are observed during training, e.g., 500 characters, the model’s recognition accuracy exhibits higher sensitivity to template selection. As the training set expands, these variations progressively diminish. When the model has been trained on all 2,755 characters, the effect of different template sets is attenuated, with deviations reduced to 0.75% on HWDB and 0.50% on Printed. A similar trend emerges in the radical zero-shot scenario, where the standard deviation gradually drops as the number of recognizable radicals increases.
> > >
> > > | Datasets | CZS-500 | CZS-1000 | CZS-1500 | CZS-2000 | CZS-2755 | RZS-50 | RZS-40 | RZS-30 | RZS-20 | RZS-10 |
> > > | :------- | :-----: | :------: | :------: | :------: | :------: | :----: | :----: | :----: | :----: | :----: |
> > > | HWDB     |  2.63   |   2.16   |   1.63   |   1.98   |   0.75   |  1.54  |  1.30  |  1.04  |  1.15  |  1.68  |
> > > | Printed  |  1.56   |   1.34   |   0.74   |   0.47   |   0.50   |  1.58  |  1.20  |  0.74  |  0.61  |  0.32  |
> > >
> > >
> > >
> > > **Details Cost/Speed**
> > >
> > > The inference time of CoLa and the comparison models on the dataset with 3,755 types of characters is shown in the following table. The results show that **CoLa has a shorter average inference time**. Similar to the radical-based and radical-based models, the inference time of CoLa will increase linearly with the size of the character set, as it needs to compare input components with the template components.
> > >
> > > | Methods | CoLa  | DenseRAN | HDE  |  SD  | CCR-CLIP |
> > > | ------- | :---: | :------: | :--: | :--: | :------: |
> > > | AIT(ms) | **9** |   1666   |  29  | 567  |    14    |
> > >
> > >
> > >
> > > **Sensitivity to K**
> > >
> > > We **chose K = 3 because this setting achieves the best accuracy** across multiple datasets. Increasing K beyond 3 does not improve performance and may lead to redundant or unstable components, while smaller K limits the expressiveness of CoLa. The results are shown in the following tables.
> > >
> > > | #Comp | HWDB-500  | HWDB-1000 | HWDB-1500 | HWDB-2000 | HWDB-2755 | Printed-500 | Printed-1000 | Printed-1500 | Printed-2000 | Printed-2755 |
> > > | :---- | :-------: | :-------: | :-------: | :-------: | :-------: | :---------: | :----------: | :----------: | :----------: | :----------: |
> > > | K = 1 |   41.19   |   39.30   |   46.28   |   47.97   |   53.82   |    78.08    |    83.24     |    87.03     |    88.05     |    85.40     |
> > > | K = 2 |   58.20   |   66.07   |   70.23   |   72.38   |   79.34   |    78.11    |    84.51     |    90.89     |    92.40     |    92.80     |
> > > | K = 3 | **68.59** | **76.58** | **79.16** | **81.16** | **82.71** |    78.10    |  **85.38**   |    90.32     |  **93.26**   |    92.70     |
> > > | K = 4 |   58.08   |   69.67   |   73.99   |   75.34   |   82.16   |    78.24    |    84.38     |    90.82     |    92.68     |  **93.61**   |
> > > | K = 5 |   55.44   |   68.85   |   71.86   |   74.32   |   81.09   |  **78.26**  |    84.24     |  **90.94**   |    92.36     |    93.29     |
> > >
> > > | #Comp |  HWDB-50  |  HWDB-40  |  HWDB-30  |  HWDB-20  |  HWDB-10  | Printed-50 | Printed-40 | Printed-30 | Printed-20 | Printed-10 |
> > > | :---- | :-------: | :-------: | :-------: | :-------: | :-------: | :--------: | :--------: | :--------: | :--------: | :--------: |
> > > | K = 1 |   39.58   |   45.77   |   55.26   |   52.00   |   49.44   |   79.79    |   80.82    |   82.79    |   87.57    |   89.22    |
> > > | K = 2 |   64.10   |   68.83   |   75.18   |   75.44   |   68.12   |   82.46    |   84.54    |   86.87    |   90.85    |   94.37    |
> > > | K = 3 | **70.40** | **74.80** |   77.01   | **80.64** | **75.78** |   82.23    |   84.48    |   82.20    | **92.12**  | **94.81**  |
> > > | K = 4 |   69.62   |   74.43   | **78.25** |   80.62   |   73.09   | **82.31**  | **85.00**  | **87.34**  |   91.96    |   94.65    |
> > > | K = 5 |   68.77   |   73.01   |   77.43   |   79.23   |   62.00   |   82.22    |   84.88    |   86.79    |   91.34    |   94.39    |

---

> > > > ### Author Response · Authors · 2025-12-02
> > > > **Response to Reviewer kKYY (Part 4)**
> > > >
> > > > **Sensitivity to Template Count N**
> > > >
> > > > We **investigate how the number of template character images influences the performance of CoLa**. The results are shown in the following tables. Across both the HWDB and Printed datasets, expanding the template set consistently improves matching accuracy, suggesting that a richer reference pool enables the model to better capture intra-class variations and reduce ambiguity in zero-shot recognition. However, the benefit of adding more templates gradually diminishes as the number increases. In particular, performance gains become less when N>10, indicating that the model has already saturated its ability to benefit from additional examples. Considering both the empirical results and the practical cost of maintaining larger template sets, we adopt N=10 templates per class, which offers a trade-off between accuracy and efficiency.
> > > >
> > > > | #Temp  | HWDB-500  | HWDB-1000 | HWDB-1500 | HWDB-2000 | HWDB-2755 | Printed-500 | Printed-1000 | Printed-1500 | Printed-2000 | Printed-2755 |
> > > > | ------ | :-------: | :-------: | :-------: | :-------: | :-------: | :---------: | :----------: | :----------: | :----------: | :----------: |
> > > > | N = 1  |   41.05   |   51.56   |   56.23   |   61.32   |   65.82   |    60.39    |    69.52     |    81.50     |    84.13     |    85.73     |
> > > > | N = 3  |   41.99   |   52.58   |   59.23   |   60.65   |   70.44   |    64.14    |    70.66     |    82.26     |    86.49     |    87.71     |
> > > > | N = 5  |   42.46   |   54.33   |   60.08   |   59.89   |   71.08   |    69.57    |    77.55     |    86.57     |    89.09     |    90.86     |
> > > > | N = 10 | **68.59** |   76.58   |   79.16   | **81.16** |   82.71   |    78.10    |    85.38     |    90.32     |    93.26     |    92.70     |
> > > > | N = 20 |   66.92   | **76.67** | **79.94** |   81.14   | **84.11** |  **80.31**  |  **86.25**   |  **91.98**   |  **93.69**   |  **94.36**   |
> > > >
> > > > | #Temp  |  HWDB-50  |  HWDB-40  |  HWDB-30  |  HWDB-20  |  HWDB-10  | Printed-50 | Printed-40 | Printed-30 | Printed-20 | Printed-10 |
> > > > | :----- | :-------: | :-------: | :-------: | :-------: | :-------: | :--------: | :--------: | :--------: | :--------: | :--------: |
> > > > | N = 1  |   48.12   |   53.88   |   58.93   |   61.04   |   63.45   |   64.26    |   69.85    |   79.45    |   83.88    |   87.00    |
> > > > | N = 3  |   50.36   |   60.19   |   63.38   |   67.01   |   67.70   |   71.25    |   72.26    |   81.61    |   85.60    |   89.96    |
> > > > | N = 5  |   53.53   |   60.70   |   63.87   |   66.91   |   61.89   |   75.52    |   77.75    |   83.55    |   88.41    |   92.45    |
> > > > | N = 10 |   70.40   |   74.80   |   77.01   |   80.64   |   75.78   |   82.23    |   84.48    |   82.20    |   92.12    |   94.81    |
> > > > | N = 20 | **74.27** | **78.07** | **81.19** | **82.70** | **83.42** | **83.43**  | **87.11**  | **91.63**  | **93.07**  | **95.36**  |
> > > >
> > > >
> > > >
> > > > **Teacher Dependence**
> > > >
> > > > The following table **summarizes the performance of CoLa when trained with different teacher encoders**, where DINOv2 outperforms CLIP under most configurations. CLIP may focus more on high-level semantic alignment and tend to downplay local structural cues. Since CoLa relies on component-level comparison, representations that retain detailed visual signals offer more effective supervision, which explains the consistent performance advantage of DINOv2 over CLIP. Fine-tuning the pretrained encoder with the character classification objective improves performance. When DINOv2 is fine-tuned on character images (DINOv2 ViT-B/16 + Ft.), the model gains additional domain-specific priors.
> > > >
> > > > | Teachers               | Finetune |  HWDB-50  |  HWDB-40  |  HWDB-30  |  HWDB-20  |  HWDB-10  |
> > > > | ---------------------- | :------: | :-------: | :-------: | :-------: | :-------: | :-------: |
> > > > | CLIP ViT-B/16          |    ✓     |   49.96   |   59.36   |   60.60   |   64.57   |   66.60   |
> > > > | DINOv2 ViT-B/16        |          |   66.33   |   72.15   |   74.09   |   76.81   | **78.29** |
> > > > | DINOv2 ViT-B/16 (CoLa) |    ✓     | **70.40** | **74.80** | **77.01** | **80.64** |   75.78   |

---

### Official Review · Reviewer_xBW8 · 2025-10-30

**Soundness:** 3
**Presentation:** 2
**Contribution:** 2
**Rating:** 4
**Confidence:** 4

**Summary:**

The paper proposed a method to decompose Chinese characters into components and then recognize them. The core contribution is a latent variable model that learns components of Chinese characters without relying on human-defined decomposition annotations. Overall, the problem addressed in the paper is interesting and the method is relatively novel, but the experimental results indicate that the method is not yet satisfactory.

**Strengths:**

1. Decomposing Chinese characters into meaningful components using unsupervised learning is an interesting topic of good academic value.
2. To my best knowledge, this is the first work that explores unsupervised component decomposition for Chinese characters. The overall framework that combines decomposition and recognition is also novel.
3. As far as I can see, the proposed method is technically sound.
4. The paper is well-organized and clearly written.

**Weaknesses:**

1. My main concern about the paper is its performance. From the visualization results (especially Fig. 6), the model frequently outputs fragmented and discontinuous components, which are inconsistent with human understanding of character components. Given that most of the demonstrated characters have simple structures (such as left-right or top-bottom), this indicates that the model has not learned to decompose characters into meaningful components.
2. The core part of the proposed method (slot attention and feature reconstruction) is directly based on an existing method ('Object-Centric Learning with Slot Attention'), which limits the novelty of the method.
3. Due to the high variability in Chinese character structures, it is crucial to either output a variable number of components based on the input character or set a relatively large K. However, the paper fixes the number of components at K=3, without any explanation or relevant experimental analysis.

**Questions:**

In the visualization experiments, it is recommended to first categorize Chinese characters by their structural types and then present the results by category, enabling readers to gain a more comprehensive understanding of the model's decomposition performance on characters with different structures.

---

> ### Author Response · Authors · 2025-12-02
> **Response to Reviewer xBW8 (Part 1)**
>
> We sincerely thank the reviewer for the constructive comments. We address the concerns as follows.
>
> **Q1. Visualization Results**
>
> > My main concern about the paper is its performance. From the visualization results (especially Fig. 6), the model frequently outputs fragmented and discontinuous components, which are inconsistent with human understanding of character components. Given that most of the demonstrated characters have simple structures (such as left-right or top-bottom), this indicates that the model has not learned to decompose characters into meaningful components.
> >
> > In the visualization experiments, it is recommended to first categorize Chinese characters by their structural types and then present the results by category, enabling readers to gain a more comprehensive understanding of the model's decomposition performance on characters with different structures.
>
> We thank the reviewer for the constructive comments. Since characters are not decomposed using human-defined radical annotations, the resulting masks may split one radical into multiple visual clusters or merge several radicals into a single region. We do not train the model to reproduce the human-defined radical decomposition scheme, but to discover generalizable latent components.
>
> Following the reviewer’s suggestion, **we added visualizations grouped by major Chinese character structures** (e.g., Left–Right, Up–Down, Surrounding structures, etc.) in Appendix D.10. These examples better illustrate how CoLa adapts component usage to different structures. We hope the additional visualization results provide a clearer and more comprehensive view of CoLa’s decomposition scheme.
>
>
>
> **Q2. The proposed method is directly based on Slot Attention**
>
> > The core part of the proposed method (slot attention and feature reconstruction) is directly based on an existing method ('Object-Centric Learning with Slot Attention'), which limits the novelty of the method.
>
> We appreciate the reviewer’s comment. Slot Attention is included in CoLa only as a component extractor, not as the core methodology. Our contribution does not lie in reusing Slot Attention, but in introducing a deep latent variable model for character decomposition, which learns self-emergent components from characters without human-defined annotations like radicals or strokes.

---

> > ### Author Response · Authors · 2025-12-02
> > **Response to Reviewer xBW8 (Part 2)**
> >
> > **Q3. Fixes the number of components**
> >
> > > Due to the high variability in Chinese character structures, it is crucial to either output a variable number of components based on the input character or set a relatively large K. However, the paper fixes the number of components at K=3, without any explanation or relevant experimental analysis.
> >
> > The hyperparameter K is only the maximum number of latent components. CoLa discovers the number of components adaptively, resulting in 1–2 components for simple characters, and up to 3 components when structural complexity increases. We chose K = 3 because this setting achieves the best accuracy across multiple datasets. Increasing K beyond 3 does not improve performance and may lead to redundant or unstable components, while smaller K limits the expressiveness of CoLa. The results are shown in the following tables.
> >
> > Table 1: Performance on the character zero-shot settings
> >
> > | #Comp | HWDB-500  | HWDB-1000 | HWDB-1500 | HWDB-2000 | HWDB-2755 | Printed-500 | Printed-1000 | Printed-1500 | Printed-2000 | Printed-2755 |
> > | :---- | :-------: | :-------: | :-------: | :-------: | :-------: | :---------: | :----------: | :----------: | :----------: | :----------: |
> > | K = 1 |   41.19   |   39.30   |   46.28   |   47.97   |   53.82   |    78.08    |    83.24     |    87.03     |    88.05     |    85.40     |
> > | K = 2 |   58.20   |   66.07   |   70.23   |   72.38   |   79.34   |    78.11    |    84.51     |    90.89     |    92.40     |    92.80     |
> > | K = 3 | **68.59** | **76.58** | **79.16** | **81.16** | **82.71** |    78.10    |  **85.38**   |    90.32     |  **93.26**   |    92.70     |
> > | K = 4 |   58.08   |   69.67   |   73.99   |   75.34   |   82.16   |    78.24    |    84.38     |    90.82     |    92.68     |  **93.61**   |
> > | K = 5 |   55.44   |   68.85   |   71.86   |   74.32   |   81.09   |  **78.26**  |    84.24     |  **90.94**   |    92.36     |    93.29     |
> >
> > Table 2: Performance on the radical zero-shot settings
> >
> > | #Comp |  HWDB-50  |  HWDB-40  |  HWDB-30  |  HWDB-20  |  HWDB-10  | Printed-50 | Printed-40 | Printed-30 | Printed-20 | Printed-10 |
> > | :---- | :-------: | :-------: | :-------: | :-------: | :-------: | :--------: | :--------: | :--------: | :--------: | :--------: |
> > | K = 1 |   39.58   |   45.77   |   55.26   |   52.00   |   49.44   |   79.79    |   80.82    |   82.79    |   87.57    |   89.22    |
> > | K = 2 |   64.10   |   68.83   |   75.18   |   75.44   |   68.12   |   82.46    |   84.54    |   86.87    |   90.85    |   94.37    |
> > | K = 3 | **70.40** | **74.80** |   77.01   | **80.64** | **75.78** |   82.23    |   84.48    |   82.20    | **92.12**  | **94.81**  |
> > | K = 4 |   69.62   |   74.43   | **78.25** |   80.62   |   73.09   | **82.31**  | **85.00**  | **87.34**  |   91.96    |   94.65    |
> > | K = 5 |   68.77   |   73.01   |   77.43   |   79.23   |   62.00   |   82.22    |   84.88    |   86.79    |   91.34    |   94.39    |

---

### Official Review · Reviewer_dEFu · 2025-10-31

**Soundness:** 3
**Presentation:** 3
**Contribution:** 1
**Rating:** 2
**Confidence:** 4

**Summary:**

This paper addresses the zero-shot Chinese Character Recognition (CCR) problem, a challenge arising from the long-tail distribution of Chinese character datasets. Inspired by human cognitive principles of compositionality and learning-to-learn, the authors propose CoLa, a deep latent variable model that learns compositional latent components of Chinese characters without relying on human-defined schemes (e.g., radicals or strokes). CoLa decomposes characters into latent components via input/template encoding, reconstructs high-level features using a frozen DINOv2 teacher encoder, and predicts classes by comparing components in the latent space. Experiments show CoLa outperforms baselines (e.g., CCR-CLIP) in character/radical zero-shot settings on HWDB, Printed, and Historical Document datasets, and generalizes to oracle bone, Japanese, and Korean characters.

**Strengths:**

Originality： Self-supervised component learning for CCR: CoLa treats components as latent variables that emerge via reconstruction and classification objectives, avoiding bias from manual decomposition rules.
Cognitive inspiration: Grounding the model in compositionality and learning-to-learn (rather than heuristics) provides a principled inductive bias, which is a thoughtful design choice for character recognition.

Rigorous experimental design: Multi-dataset evaluation (handwritten, printed, historical) and ablation studies (teacher encoder, prediction loss) confirm model components’ necessity. Quantitative (mIoU for component masks) and qualitative (visualization) analyses of interpretability add rigor.

**Weaknesses:**

Primary Weakness: Insufficient Topic Significance and Practical Relevance
The most critical limitation of this work is that the topic of "zero-shot Chinese character recognition" lacks broad practical demand and generalizable research value, making the contribution narrow and marginal in the broader CV landscape.

The zero-shot scenarios CoLa targets (e.g., training on 500 characters to recognize 1,000 rare characters, radical zero-shot with n<10) are artificially extreme and rarely encountered in practice:

Chinese characters are highly structured, low-diversity visual objects (fixed writing norms, finite structural patterns) that differ fundamentally from natural images (e.g., scene understanding, object detection) or unstructured text (e.g., handwritten Latin script). CoLa’s core innovation—self-learning compositional latent components—relies on the unique structure of Chinese characters and cannot be transferred to other CV tasks:

Lack of Comparison with Practical Alternatives： The paper ignores simpler, more practical solutions for rare character recognition that outperform CoLa in real-world scenarios:

Overemphasis on Niche Cross-Dataset Generalization

CoLa’s cross-dataset results (OBCs, Japanese, Korean) are technically interesting but further highlight the topic’s marginality:
Japanese (with hiragana/katakana) and Korean (with hangul) have distinct writing systems—CoLa’s ability to decompose their characters is irrelevant to their core recognition challenges (e.g., hangul’s syllabic structure).

No quantitative metrics (e.g., retrieval accuracy for OBCs/Japanese) are provided, making it impossible to assess whether this generalization has practical value for researchers in these fields.

Few-shot learning (e.g., fine-tuning CCR-CLIP on 10 samples per rare character) would likely achieve comparable or higher accuracy than CoLa’s zero-shot approach, with lower computational cost.

Human-in-the-loop annotation (common in 古籍整理) is more efficient for rare characters, as experts can label 100 samples in hours—rendering zero-shot models unnecessary.

**Questions:**

Topic Significance Justification:

The authors focus on zero-shot Chinese character recognition, but this problem is niche with limited real-world demand. Could you:
Provide concrete examples of industrial or academic users who require zero-shot recognition of rare characters (e.g., training 500 characters to recognize 1,000 rare ones)?

Explain why this problem is more important than addressing core CCR pain points (e.g., low-resolution character recognition, scene text with complex backgrounds) or broader CV challenges (e.g., general zero-shot object detection)?

Generalizability and Transfer Value:

CoLa’s component learning is tailored to Chinese characters. Can you demonstrate that your method (or its core ideas) can be transferred to other CV tasks (e.g., decomposing natural objects like "car" or "cat," or recognizing unstructured handwritten Latin text)?

Practical Alternative Comparison:

Few-shot learning or human annotation is more cost-effective for rare character recognition. Have you compared CoLa with few-shot baselines (e.g., CCR-CLIP fine-tuned on 5–10 samples per rare character)? If CoLa is outperformed by these simpler methods in real-world rare character scenarios, what is its competitive advantage?

---

> ### Author Response · Authors · 2025-12-02
> **Response to Reviewer dEFu (Part 1)**
>
> We sincerely thank the reviewer for the constructive comments. We address the concerns as follows.
>
> **Q1. Insufficient Topic Significance and Practical Relevance.**
>
> > The most critical limitation of this work is that the topic of "zero-shot Chinese character recognition" lacks broad practical demand and generalizable research value, making the contribution narrow and marginal in the broader CV landscape.
> >
> > Explain why this problem is more important than addressing core CCR pain points (e.g., low-resolution character recognition, scene text with complex backgrounds) or broader CV challenges (e.g., general zero-shot object detection)?
>
> We appreciate the reviewer’s concern and would like to clarify that the core topic of **our work is learning compositional components from Chinese characters**, rather than solving the zero-shot Chinese character recognition task. Zero-shot CCR is a downstream task that demonstrates whether CoLa has learned reusable latent components.
>
> We do **not** claim that zero-shot CCR is “more important” than other OCR problems, such as low-resolution character recognition and scene text with complex backgrounds. These research directions are complementary rather than mutually exclusive.
>
> The practical value of zero-shot CCR is significant in scenarios where supervised labels are absent, such as rare characters in historical documents, variant forms, or ancient scripts like oracle bone inscriptions that remain partially undeciphered. Compositional decomposition will help recognition in these cases by transferring the learned character decomposition scheme to unseen characters.
>
>
>
> **Q2. Overemphasis on Niche Cross-Dataset Generalization**
>
> > The zero-shot scenarios CoLa targets (e.g., training on 500 characters to recognize 1,000 rare characters, radical zero-shot with n<10) are artificially extreme and rarely encountered in practice:
> >
> > The authors focus on zero-shot Chinese character recognition, but this problem is niche with limited real-world demand. Could you: Provide concrete examples of industrial or academic users who require zero-shot recognition of rare characters (e.g., training 500 characters to recognize 1,000 rare ones)?
>
> We appreciate the reviewer’s concern about the limited experimental setup. We would like to clarify that the experimental setup (e.g., training on 500 characters and recognizing 1,000 unseen ones) is **not intended as an artificial extreme**. It serves as a controlled way to test whether a model has genuinely learned components that support compositional generalization.
>
> Such zero-shot conditions are not hypothetical. In oracle bone characters, around 4,500 distinct character classes have been discovered, but only about 2,200 have been reliably deciphered [1]. The remaining over 2,000 characters currently lack confirmed annotations or labels. In this case, we can hardly “train on all categories” but must infer unseen or partially understood characters from known components and structures, which is close to our experimental design.
>
> [1] Li, Jing, et al. "A comprehensive survey of oracle character recognition: challenges, benchmarks, and beyond."

---

> > ### Author Response · Authors · 2025-12-02
> > **Response to Reviewer dEFu (Part 2)**
> >
> > **Q3. Generalization to other CV tasks**
> >
> > > Chinese characters are highly structured, low-diversity visual objects (fixed writing norms, finite structural patterns) that differ fundamentally from natural images (e.g., scene understanding, object detection) or unstructured text (e.g., handwritten Latin script). CoLa’s core innovation—self-learning compositional latent components—relies on the unique structure of Chinese characters and cannot be transferred to other CV tasks:
> > >
> > > CoLa’s component learning is tailored to Chinese characters. Can you demonstrate that your method (or its core ideas) can be transferred to other CV tasks (e.g., decomposing natural objects like "car" or "cat," or recognizing unstructured handwritten Latin text)?
> >
> > The idea of decomposing a target into meaningful parts is a general principle. A component is not necessarily a radical or a stroke cluster, but any latent visual substructure that reliably explains the object’s variation. **The compositional latent components can be extended to other vision tasks**, e.g., a dog can be decomposed into ears, muzzle, limbs, etc. The compositional mechanism in CoLa is not hard-coded to Chinese characters. CoLa is encouraged to discover internally reusable substructures, which also applies to natural objects.
> >
> > We have evaluated CoLa on the fine-grained visual categorization benchmark Stanford Dogs [1], which contains about 20,000 real-world dog images with small inter-class differences. Approximately 10 out of 120 classes are held out as unseen test classes to construct a zero-shot categorization setting. The results are shown in the following table.
> >
> > | Models   | Random Baseline | CLIP | DINOv2 | Slot Attention |   CoLa    |
> > | :------- | :-------------: | :--: | :----: | :------------: | :-------: |
> > | Accuracy |      0.83       | 0.01 |  7.72  |      7.34      | **22.26** |
> >
> > Since the compared CCR method cannot be directly applied to the benchmark, we compare CoLa with three representative models: CLIP, DINOv2, and Slot Attention. The results show that **CoLa achieves the best performance**, with CLIP’s performance even falling below the random baseline. This contrast highlights an important limitation of CLIP. The CLIP features tend to emphasize high-level semantic concepts (i.e., recognizing that all samples are “dogs”), while ignoring the fine-grained information to distinguish different dog categories. The compositional latent components of CoLa can capture variations between dogs for zero-shot classification on this fine-grained dataset.
> >
> > [1] Khosla, Aditya, et al. "Novel dataset for fine-grained image categorization: Stanford dogs."
> >
> >
> > **Q4. Comparsion to few-shot models**
> >
> > > Few-shot learning (e.g., fine-tuning CCR-CLIP on 10 samples per rare character) would likely achieve comparable or higher accuracy than CoLa’s zero-shot approach, with lower computational cost.
> > >
> > > Human-in-the-loop annotation is more efficient for rare characters, as experts can label 100 samples in hours—rendering zero-shot models unnecessary.
> > >
> > > Few-shot learning or human annotation is more cost-effective for rare character recognition. Have you compared CoLa with few-shot baselines (e.g., CCR-CLIP fine-tuned on 5–10 samples per rare character)? If CoLa is outperformed by these simpler methods in real-world rare character scenarios, what is its competitive advantage?
> >
> > We thank the reviewer for this feedback. Few-shot fine-tuning can hardly address the difficulties of recognizing rare Chinese characters in historical documents, e.g., oracle bone characters (OBCs) do not have readily accessible labels. Annotating these characters usually involves paleographic research and semantic interpretation. Though 4,500 OBC character classes have been identified, only about 2,200 have been reliably deciphered, meaning that more than half of the known categories still lack stable annotation. In such domains, it is hard to collect labeled samples per class.
> >
> > The competitive advantage of CoLa is **not** to outperform every possible supervised method when labels are abundant, but to provide recognition capability when labels are unavailable or incomplete. CoLa operates through decomposition and component matching, enabling recognition of genuinely unseen categories without re-training or per-class annotation. This capability is complementary to human-in-the-loop or few-shot approaches. CoLa can assist experts in identifying unknown forms and offers a meaningful starting point in domains where full supervision is unattainable.

---

> > > ### Author Response · Authors · 2025-12-02
> > > **Response to Reviewer dEFu (Part 3)**
> > >
> > > **Q5. Concern about the cross-dataset results**
> > >
> > > > CoLa’s cross-dataset results (OBCs, Japanese, Korean) are technically interesting but further highlight the topic’s marginality: Japanese (with hiragana/katakana) and Korean (with hangul) have distinct writing systems—CoLa’s ability to decompose their characters is irrelevant to their core recognition challenges (e.g., hangul’s syllabic structure).
> > >
> > > We thank the reviewer for raising this point. Our cross-dataset experiments are **not** intended to claim that CoLa solves the linguistic challenges of Japanese or Korean writing systems. They serve to validate whether the proposed compositional latent components can generalize across distinct domains. The objective of CoLa is learning components that can explain the characters. And the cross-dataset results provide evidence that CoLa is not restricted to a specific writing system and that the decomposition ability remains meaningful in other domains.
> > >
> > >
> > >
> > > **Q6. Quantitative metrics for OBCs/Japanese**
> > >
> > > > No quantitative metrics (e.g., retrieval accuracy for OBCs/Japanese) are provided, making it impossible to assess whether this generalization has practical value for researchers in these fields.
> > >
> > > To assess the model performance on the Japanese, Korean, and OBC datasets, **we employ several metrics that are commonly used in zero-shot retrieval tasks**. Recall@K measures whether at least one correct class appears within the top-K matched templates, and is therefore an appropriate indicator of match success in the zero-shot setting. Prec@K quantifies the proportion of correctly matched templates among the top-K results, which is informative when multiple examples exist for each class. To jointly capture both aspects, we further report F1@K, which is the harmonic mean of Prec@K and Recall@K. In addition, we employ the Mean Reciprocal Rank (MRR) to evaluate how early the correct class appears in the ranked candidates. The following tables present the quantitative results of different cross-script character recognition configurations. Across the configurations, we observe that CoLa outperforms other baselines, demonstrating its generalization ability in different writing systems.
> > >
> > >
> > >
> > > Table 1: Retrieval accuracy on Japanese
> > >
> > > | Models         | Recall@1  | Recall@5  |  Prec@1   |  Prec@5   |   F1@1    |   F1@5    |    MRR    |
> > > | :------------- | :-------: | :-------: | :-------: | :-------: | :-------: | :-------: | :-------: |
> > > | CLIP           |   0.67    |   3.36    |   0.67    |   0.67    |   0.67    |   1.12    |   10.45   |
> > > | DINOv2         |   64.07   |   90.65   |   64.07   |   53.49   |   64.07   |   63.84   |   75.79   |
> > > | DINOv2 Ft.     |   79.01   |   96.47   |   79.01   |   70.80   |   79.01   |   78.55   |   86.64   |
> > > | Slot Attention |   76.67   |   95.52   |   76.67   |   71.42   |   76.67   |   78.42   |   84.85   |
> > > | **CoLa**       | **83.03** | **97.95** | **83.03** | **77.77** | **83.03** | **83.99** | **89.62** |
> > >
> > > Table 2: Retrieval accuracy on Korean
> > >
> > > | Models         | Recall@1  | Recall@5  |  Prec@1   |  Prec@5   |   F1@1    |   F1@5    |    MRR    |
> > > | :------------- | :-------: | :-------: | :-------: | :-------: | :-------: | :-------: | :-------: |
> > > | CLIP           |   0.07    |   0.33    |   0.07    |   0.07    |   0.07    |   0.11    |   9.22    |
> > > | DINOv2         |   3.98    |   10.17   |   3.98    |   2.27    |   3.98    |   3.66    |   14.56   |
> > > | DINOv2 Ft.     |   11.97   |   30.61   |   11.97   |   8.56    |   11.97   |   12.91   |   25.34   |
> > > | Slot Attention |   13.42   |   30.60   |   13.42   |   8.05    |   13.42   |   12.36   |   26.19   |
> > > | **CoLa**       | **15.03** | **39.58** | **15.03** | **11.71** | **15.03** | **17.40** | **29.89** |
> > >
> > > Table 3: Retrieval accuracy on OBC
> > >
> > > | Models         | Recall@1  | Recall@5  |  Prec@1   |  Prec@5   |   F1@1    |   F1@5    |    MRR    |
> > > | :------------- | :-------: | :-------: | :-------: | :-------: | :-------: | :-------: | :-------: |
> > > | CLIP           |   0.01    |   0.06    |   0.01    |   0.01    |   0.01    |   0.02    |   9.12    |
> > > | DINOv2         |   49.78   |   65.88   |   49.78   |   25.54   |   49.78   |   34.84   |   59.26   |
> > > | DINOv2 Ft.     |   53.80   |   69.68   |   53.80   |   28.60   |   53.80   |   38.32   |   62.83   |
> > > | Slot Attention |   45.97   |   59.30   |   45.97   |   21.83   |   45.97   |   30.19   |   54.87   |
> > > | **CoLa**       | **61.97** | **74.84** | **61.97** | **33.67** | **61.97** | **43.88** | **69.39** |

---

### Official Review · Reviewer_kd4t · 2025-11-01

**Soundness:** 3
**Presentation:** 3
**Contribution:** 2
**Rating:** 6
**Confidence:** 3

**Summary:**

This paper introduces CoLa (Compositional Latent components), a deep latent variable model designed to recognize Chinese characters, especially in zero-shot scenarios where characters seen during testing were absent from the training data.

Unlike previous methods that rely on rigid, human-defined decomposition rules (like predefined strokes or radicals) , CoLa learns to automatically discover its own compositional components as latent variables. This self-learning approach allows it to generalize far better to unseen characters and character parts.

**Strengths:**

The paper's main strength is moving beyond rigid, human-defined decomposition schemes (like strokes or radicals). The CoLa model instead learns to decompose characters on its own, which is inspired by the cognitive principles of "compositionality" and "learning-to-learn"

The visualizations show that the model is not just a black box. The learned latent components (Cmp#1, #2, #3) clearly focus on distinct, independent regions of the character.

Instead of trying to reconstruct noisy, raw pixels, the model is trained to reconstruct the high-level semantic features from a frozen, pre-trained DINOv2 encoder. An ablation study confirms this "teacher" is "crucial" for the model's success

**Weaknesses:**

The model architecture hard-codes the number of components to K=3

The model's success relies on a DINOv2 "teacher" that was pre-trained on natural images (like animals, objects, and landscapes). While this works, its features are not optimized for the specific domain of orthography.

**Questions:**

Since DINOv2 was trained on general-domain natural images (animals, landscapes, etc.), is the model learning to decompose characters into "natural image parts" (e.g., generic lines and curves) rather than "orthographic parts"?

How sensitive is the model to this template set?

---

> ### Author Response · Authors · 2025-12-02
> **Response to Reviewer kd4t (Part 1)**
>
> We sincerely thank the reviewer for the constructive comments. We address the concerns as follows.
>
> **Q1. The model architecture hard-codes the number of components to K=3**
>
> The number of latent components is not hard-coded as a semantic assumption, but selected as a capacity upper bound for latent components. **K=3 does not force the model to decompose characters into three parts.** CoLa discovers the number of components adaptively, resulting in 1–2 components for simple characters, and up to 3 components when structural complexity increases.
>
> We chose K = 3 because this setting achieves the best accuracy across multiple datasets. Increasing K beyond 3 does not improve performance and may lead to redundant or unstable components, while smaller K limits the expressiveness of CoLa. The results are shown in the following tables.
>
> Table 1: Performance on the character zero-shot settings
>
> | #Comp | HWDB-500  | HWDB-1000 | HWDB-1500 | HWDB-2000 | HWDB-2755 | Printed-500 | Printed-1000 | Printed-1500 | Printed-2000 | Printed-2755 |
> | :---- | :-------: | :-------: | :-------: | :-------: | :-------: | :---------: | :----------: | :----------: | :----------: | :----------: |
> | K = 1 |   41.19   |   39.30   |   46.28   |   47.97   |   53.82   |    78.08    |    83.24     |    87.03     |    88.05     |    85.40     |
> | K = 2 |   58.20   |   66.07   |   70.23   |   72.38   |   79.34   |    78.11    |    84.51     |    90.89     |    92.40     |    92.80     |
> | K = 3 | **68.59** | **76.58** | **79.16** | **81.16** | **82.71** |    78.10    |  **85.38**   |    90.32     |  **93.26**   |    92.70     |
> | K = 4 |   58.08   |   69.67   |   73.99   |   75.34   |   82.16   |    78.24    |    84.38     |    90.82     |    92.68     |  **93.61**   |
> | K = 5 |   55.44   |   68.85   |   71.86   |   74.32   |   81.09   |  **78.26**  |    84.24     |  **90.94**   |    92.36     |    93.29     |
>
> Table 2: Performance on the radical zero-shot settings
>
> | #Comp |  HWDB-50  |  HWDB-40  |  HWDB-30  |  HWDB-20  |  HWDB-10  | Printed-50 | Printed-40 | Printed-30 | Printed-20 | Printed-10 |
> | :---- | :-------: | :-------: | :-------: | :-------: | :-------: | :--------: | :--------: | :--------: | :--------: | :--------: |
> | K = 1 |   39.58   |   45.77   |   55.26   |   52.00   |   49.44   |   79.79    |   80.82    |   82.79    |   87.57    |   89.22    |
> | K = 2 |   64.10   |   68.83   |   75.18   |   75.44   |   68.12   |   82.46    |   84.54    |   86.87    |   90.85    |   94.37    |
> | K = 3 | **70.40** | **74.80** |   77.01   | **80.64** | **75.78** |   82.23    |   84.48    |   82.20    | **92.12**  | **94.81**  |
> | K = 4 |   69.62   |   74.43   | **78.25** |   80.62   |   73.09   | **82.31**  | **85.00**  | **87.34**  |   91.96    |   94.65    |
> | K = 5 |   68.77   |   73.01   |   77.43   |   79.23   |   62.00   |   82.22    |   84.88    |   86.79    |   91.34    |   94.39    |
>
>
>
> **Q2. Question about the teacher**
>
> > The model's success relies on a DINOv2 "teacher" that was pre-trained on natural images (like animals, objects, and landscapes). While this works, its features are not optimized for the specific domain of orthography.
> >
> > Since DINOv2 was trained on general-domain natural images (animals, landscapes, etc.), is the model learning to decompose characters into "natural image parts" (e.g., generic lines and curves) rather than "orthographic parts"?
>
> The DINOv2 backbone is not used in its original checkpoint. **We finetune DINOv2 on the training character set** using a character classification objective, so the resulting visual features are optimized toward orthographic discrimination rather than generic natural-image semantics. In the following table, we provide the accuracy of CoLa using the original and finetuned DINOv2. This finetuning improves the recognition performance, indicating that the DINOv2 features shift from general visual cues to features aligned with stroke patterns, radical geometry, and compositional layouts.
>
> | Teachers               | Finetune |  HWDB-50  |  HWDB-40  |  HWDB-30  |  HWDB-20  |  HWDB-10  |
> | ---------------------- | :------: | :-------: | :-------: | :-------: | :-------: | :-------: |
> | DINOv2 ViT-B/16        |          |   66.33   |   72.15   |   74.09   |   76.81   | **78.29** |
> | DINOv2 ViT-B/16 (CoLa) |    ✓     | **70.40** | **74.80** | **77.01** | **80.64** |   75.78   |

---

> > ### Author Response · Authors · 2025-12-02
> > **Response to Reviewer kd4t (Part 2)**
> >
> > **Q3. How sensitive is the model to this template set?**
> >
> > The following table reports the standard deviation of recognition accuracy obtained using five independently constructed template sets. The results show that the variance across different template sets is influenced by the size of the training character set. **The overall influence is limited, especially when the training character set is large.** When only a small number of classes are observed during training, e.g., 500 characters, the model’s recognition accuracy exhibits higher sensitivity to template selection. As the training set expands, these variations progressively diminish. When the model has been trained on all 2,755 characters, the effect of different template sets is attenuated, with deviations reduced to 0.75% on HWDB and 0.50% on Printed. A similar trend emerges in the radical zero-shot scenario, where the standard deviation gradually drops as the number of recognizable radicals increases.
> >
> > | Datasets | CZS-500 | CZS-1000 | CZS-1500 | CZS-2000 | CZS-2755 | RZS-50 | RZS-40 | RZS-30 | RZS-20 | RZS-10 |
> > | :------- | :-----: | :------: | :------: | :------: | :------: | :----: | :----: | :----: | :----: | :----: |
> > | HWDB     |  2.63   |   2.16   |   1.63   |   1.98   |   0.75   |  1.54  |  1.30  |  1.04  |  1.15  |  1.68  |
> > | Printed  |  1.56   |   1.34   |   0.74   |   0.47   |   0.50   |  1.58  |  1.20  |  0.74  |  0.61  |  0.32  |

---

### Author Response · Authors · 2025-12-03
**Rebuttal Summary**

We sincerely thank the reviewers and the AC for their thoughtful feedback and careful evaluation. The detailed comments greatly helped us strengthen the clarity, completeness, and empirical depth of the paper. Many of the insights raised in the reviews directly motivated substantial improvements in both writing and experiments.

To address the reviewers’ concerns comprehensively, we implemented a series of revisions and additions across the manuscript:

1. We clarified that the goal of CoLa is to learn generalizable and reusable latent components from Chinese characters. Zero-shot recognition is one application scenario to demonstrate the utility of these latent components, not the core problem. We further explained why zero-shot settings naturally arise in real-world Chinese character recognition applications. (Reviewers dEFu and kKYY)
2. We explained that Slot Attention serves only as a component extraction module in CoLa. The main contributions lie in the proposed self-supervised framework of Chinese character decomposition, which is distinct from standard Slot Attention pipelines. (Reviewer xBW8)
3. We added extensive experiments evaluating different choices of K, and demonstrated that K=3 provides the best trade-off across datasets. (Reviewers kd4t, xBW8, and kKYY)
4. We added experiments comparing CLIP and DINOv2 backbones and show that fine-tuning DINOv2 leads to better component-based matching performance. (Reviewers kd4t and kKYY)
5. We added results on how different template sets and template quantities influence model performance. (Reviewers kd4t and kKYY)
6. We evaluated CoLa on the Stanford Dogs dataset to verify that the learned components are not limited to Chinese characters. (Reviewer dEFu)
7. We provided additional results on cross-style (between handwritten and printed) and cross-script scenarios (from Chinese to Japanese/Korean/OBC). (Reviewers dEFu and kKYY)
8. We added more visualizations and organized them by structures (e.g., left-right and up-down), allowing a clearer understanding of component decomposition behaviors. (Reviewer xBW8)
9. We included additional runtime analysis to compare inference efficiency with other baselines. (Reviewer kKYY)

The additional results have been included in the revised manuscript, highlighted in blue. We appreciate the reviewers’ insightful suggestions and the AC’s effort in the review process. The feedback significantly improved the presentation and empirical rigor of our work. We hope the revised manuscript now effectively resolves the reviewers' concerns.

---

### Meta-Review · Area_Chair_AhPv · 2026-01-05

**Summary:**

The submission received low scores (6, 2, 4, 4) with no strong advocacy. While the unsupervised approach is interesting, reviewers identified critical weaknesses regarding the niche nature of the problem and limited methodological novelty, noting the method is largely an application of existing architectures like Slot Attention.

**Reviewer Concerns:**

The authors provided a substantial amount of new experimental data in the rebuttal, including hyperparameter justifications (K=3) and cross-script generalization tests.

Despite the additional data, fundamental issues remain on technical novelty and arguably niche nature of the problem. Furthermore, the learned components were criticized for being visually fragmented, undermining the core claim of interpretable decomposition.

**Reviewer Scores:**

Reviewer dEFu is likely to maintain their strong opposition, as the rebuttal's defense of historical scripts can be weak. Similarly, Reviewer xBW8 would likely retain their score due to technical novelty. While Reviewers kd4t and kKYY might view the new experimental data favorably, these incremental improvements are insufficient to overcome the fundamental consensus.

---

### Decision · Program_Chairs · 2026-01-26

Reject